# Factors Modulating COVID-19: A Mechanistic Understanding Based on the Adverse Outcome Pathway Framework

**DOI:** 10.3390/jcm11154464

**Published:** 2022-07-31

**Authors:** Laure-Alix Clerbaux, Maria Cristina Albertini, Núria Amigó, Anna Beronius, Gillina F. G. Bezemer, Sandra Coecke, Evangelos P. Daskalopoulos, Giusy del Giudice, Dario Greco, Lucia Grenga, Alberto Mantovani, Amalia Muñoz, Elma Omeragic, Nikolaos Parissis, Mauro Petrillo, Laura A. Saarimäki, Helena Soares, Kristie Sullivan, Brigitte Landesmann

**Affiliations:** 1European Commission, Joint Research Centre (JRC), 21027 Ispra, Italy; sandra.coecke@ec.europa.eu (S.C.); evangelos.daskalopoulos@ec.europa.eu (E.P.D.); nikolaos.parissis@ec.europa.eu (N.P.); mauro.petrillo@seidor.com (M.P.); brigitte.landesmann@ext.ec.europa.eu (B.L.); 2Department of Biomolecular Sciences, University of Urbino Carlo Bo, 61029 Urbino, Italy; maria.albertini@uniurb.it; 3Biosfer Teslab SL., 43204 Reus, Spain; namigo@biosferteslab.com; 4Department of Basic Medical Sciences, Universitat Rovira i Virgili (URV), 23204 Reus, Spain; 5Centro de Investigación Biomédica en Red de Diabetes y Enfermedades Metabólicas Asociadas (CIBERDEM), Instituto de Salud Carlos III (ISCIII), 28029 Madrid, Spain; 6Institute of Environmental Medicine, Karolinska Institutet, 17177 Stockholm, Sweden; anna.beronius@ki.se; 7Impact Station, 1223 JR Hilversum, The Netherlands; gillina@impactstation.nl; 8Department of Pharmaceutical Sciences, Utrecht Institute for Pharmaceutical Sciences, Utrecht University, 3584 CG Utrecht, The Netherlands; 9Finnish Hub for Development and Validation of Integrated Approaches (FHAIVE), Faculty of Medicine and Health Technology, Tampere University, 33100 Tampere, Finland; giusy.delgiudice@tuni.fi (G.d.G.); dario.greco@tuni.fi (D.G.); laura.saarimaki@tuni.fi (L.A.S.); 10Département Médicaments et Technologies pour la Santé (DMTS), Université Paris-Saclay, CEA, INRAE, SPI, F-30200 Bagnols-sur-Ceze, France; lucia.grenga@cea.fr; 11Department of Food Safety, Nutrition and Veterinary Public Health, Istituto Superiore di Sanità, 00161 Rome, Italy; alberto.mantovani@iss.it; 12European Commission, Joint Research Centre (JRC), 2440 Geel, Belgium; amalia.munoz-pineiro@ec.europa.eu; 13Faculty of Pharmacy, University of Sarajevo, 71000 Sarajevo, Bosnia and Herzegovina; elma.omeragic@ffsa.unsa.ba; 14Laboratory of Immunobiology and Pathogenesis, Chronic Diseases Research Centre, Faculdade de Ciências Médicas Medical School, University of Lisbon, 1649-004 Lisbon, Portugal; helena.soares@nms.unl.pt; 15Physicians Committee for Responsible Medicine, Washington, DC 20016, USA; ksullivan@pcrm.org

**Keywords:** SARS-CoV-2 infection, COVID-19, modulating factors, adverse outcome pathway, sex, age, co-morbidities, lifestyle, environment, pre-existing conditions

## Abstract

Addressing factors modulating COVID-19 is crucial since abundant clinical evidence shows that outcomes are markedly heterogeneous between patients. This requires identifying the factors and understanding how they mechanistically influence COVID-19. Here, we describe how eleven selected factors (age, sex, genetic factors, lipid disorders, heart failure, gut dysbiosis, diet, vitamin D deficiency, air pollution and exposure to chemicals) influence COVID-19 by applying the Adverse Outcome Pathway (AOP), which is well-established in regulatory toxicology. This framework aims to model the sequence of events leading to an adverse health outcome. Several linear AOPs depicting pathways from the binding of the virus to ACE2 up to clinical outcomes observed in COVID-19 have been developed and integrated into a network offering a unique overview of the mechanisms underlying the disease. As SARS-CoV-2 infectibility and ACE2 activity are the major starting points and inflammatory response is central in the development of COVID-19, we evaluated how those eleven intrinsic and extrinsic factors modulate those processes impacting clinical outcomes. Applying this AOP-aligned approach enables the identification of current knowledge gaps orientating for further research and allows to propose biomarkers to identify of high-risk patients. This approach also facilitates expertise synergy from different disciplines to address public health issues.

## 1. Introduction

In March 2020, the World Health Organization declared the first pandemic caused by a coronavirus [1]. Researchers worldwide have intensively investigated the biological mechanisms underlying the disease and the factors rendering some populations more vulnerable, resulting in a flood of publications. In the summer of 2020, the CIAO project (Modelling the Pathogenesis of COVID-19 using the Adverse Outcome Pathway Framework) was initiated to provide a unique overview of the available knowledge on COVID-19 pathogenesis using the Adverse Outcome Pathway (AOP) framework [2,3]. The AOP concept provides a pragmatic means for organizing fast-evolving scientific knowledge based on existing data and literature from different fields [2,4]. An AOP describes a sequence of biological events starting from an initial interaction at the molecular level, called Molecular Initiating Event (MIE), through key biological events up to an adverse outcome (AO) (Figure 1) [5,6,7]. AOPs are not intended to describe all details, but rather capture essential events, called Key Events (KEs), which drive downstream KEs and ultimately the AO. Importantly, KEs must be measurable in laboratory or clinical settings. A significant distinguishing aspect of the AOP approach is the structured evaluation and representation of the weight of evidence supporting the causality of the relationships between two KEs, called Key Event Relationships (KERs) [8]. Available qualitative and quantitative information is considered in the KER weight of evidence evaluation, including the magnitude, direction, and time concordance of a perturbation in the upstream KE needed to elicit a detectable change in the downstream KE. Thus, a single AOP proposes one biological pathway leading to an AO and a set of methods to measure KEs and predict the AO. The Organisation for Economic Co-operation and Development (OECD) maintains an online platform called AOP-Wiki [9], where information captured in AOPs is openly accessible to interested researchers, regulators, and clinicians. It is important to mention that there is a rigorous approach to collect evidence at the core of AOP development; however, this approach does not rely integrally on systematic review principles.

AOPs are widely acknowledged as valuable tools in chemical safety assessments and regulatory toxicology analyses to provide methods and predictive models [10] while minimizing animal testing. The utility of AOPs for regulatory application is defined by the confidence with which AOPs facilitate extrapolation of data measured at low levels of biological organization (molecular and cellular-based methods) to relevant outcomes in organs, individuals, or populations. AOPs can also be used for assessing the mechanistic plausibility of epidemiological observations, such as the relationships between exposure to chemicals or nanomaterials and risk for disease [11]. While currently mainly exploited in toxicology, AOPs could have great value for other fields, such as biomedical research, thus profiting from growing AOP networks that describe the effects of chemical and non-chemical stressors. Human diseases are generally classified based on clinical phenotypes. Transitioning to a mechanistically-based understanding of the diseases could facilitate the design of hypothesis-based research strategies, improve predictive modeling and contribute to drug discovery [8,12]. 

The CIAO project explores this broader application of AOPs by expanding it to a viral stressor, namely SARS-CoV-2 [2,3,13]. Within the project, many linear AOPs have been developed describing key steps leading to a specific AO in COVID-19 [14]. The AOP framework also considers modulating factors (MFs) that intervene at KERs, affecting the quantitative response-response relationship between two Kes [15]. Addressing MFs is crucial since abundant clinical evidence shows that COVID-19 outcomes are markedly heterogeneous. Patients range from being asymptomatic or having a mild upper respiratory illness to having severe lung injury that requires hospitalization and may progress to hyperinflammation, acute respiratory distress syndrome, multi-organ failure, and death [16,17]. The variation in clinical outcomes suggests that socio-demographic and biological factors, characteristic of individuals or populations, modulate the course of the disease, corresponding to the concept of the epidemiological triad in infectious diseases. A better understanding of these host-virus interactions could allow for the refinement of preventive measures and improvement of treatment options. This requires identifying MFs contributing to outcome heterogeneity and a mechanistic understanding of their influence on COVID-19. Here, we specifically focus on linking eleven selected MFs to initial and central mechanisms underlying COVID-19 pathogenesis in an AOP-aligned approach. We showcase how this knowledge management framework can help to structure mechanistic data in the context of an infectious disease, reveal how different risk factors influence specific biological processes impacting the course of the disease, and show the interplay between various MFs.

## 2. Factors Modulating COVID-19: Epidemiological and Clinical Data

While there is a multitude of factors modulating COVID-19, eleven were investigated in this study, representing a non-exhaustive list. These MFs were selected based on epidemiological and/or clinical studies to be representative of different categories: intrinsic (age, sex and genetic factors), co-morbidities (history of dyslipidemia, obesity, pre-existing heart failure and gut dysbiosis), lifestyle-related (vitamin D deficiency and diet) and environmental (air pollution and exposure to chemicals) (Figure 2). Socioeconomic, psychological or cultural aspects might also influence the onset and severity of the disease but will not be discussed here.

### 2.1. Biological (Intrinsic)

**Age**. Age is by far the strongest risk factor for life-threatening COVID-19. Age distribution of COVID-19 deaths has been supported by multivariate analyses across continents [18,19]. The infection fatality ratio has been estimated to be lowest among 5- to 9-year-old children, with a log-linear increase in individuals older than 30 years [20]. Most deaths involved people older than 80 years, in particular in care facilities [21,22,23,24], while people over 65 years without co-morbidities have a very low COVID-19 mortality rate [25]. Surprisingly, infants and young toddlers seem protected from severe disease [26].

**Sex.** Epidemiological data and observational reports have shown that COVID-19 has different outcomes for men and women. Despite similar infection rates in men and women, men are at higher risk for hospitalisation, admission to an intensive care unit (ICU), and death [27,28]. COVID-19 outcomes are also influenced by gender differences, the different social roles and behavioural factors, but these aspects are not discussed here [29].

**Genetic factors**. Undoubtedly, host genetic factors also play a role in SARS-CoV-2 pathophysiology, influencing an individual or a population susceptibility [30,31]. Here, we aim to investigate the impact on COVID-19 of polymorphisms in **ACE2 and TMPRSS2**, serving as main SARS-CoV-2 cell entry gateways, as well as in Toll-like receptors (TLRs), acting as crucial actors of the immune system alert during infections. Epidemiological studies showed that around twenty natural ACE2 variants might partially account for the differences of COVID-19 prevalence and mortality rates observed between Europe and East Asia [32,33]. Regarding **TLRs**, it has been shown that certain patients with severe COVID-19 were associated with a rare putative loss-of-function variants of X-chromosomal TLR7, that causes poor defense against coronavirus [34,35]. Gene analysis also revealed elevated expression of TLR7 and TLR9, leading to exaggerated immune response, in certain ethnic groups with higher COVID-19 mortality [36,37]. Clinical evidence showing associations between TLR gain-of-function polymorphisms and higher incidence of ICU acquired infection have been reported for other immune-mediated diseases and infections [38,39,40]. In addition, patients with **blood group A** were reported to have an increased risk of severe COVID-19 while the protective effect of the O allele is small, with an odds ratio of ~0.90.

### 2.2. Pre-Existing Co-Morbidities

**Atherogenic dyslipidemia and obesity.** Beyond age, sex, and genetic factors, co-morbidities are the main clinical determinants of COVID-19 severity [17,41,42,43]. Particularly, COVID-19 patients with metabolic syndrome had significantly higher hospitalization and mortality rates [44,45]. Metabolic syndrome is the common denominator of hypertension, diabetes, obesity and dyslipidemia. Here, we will particularly investigate how a history of dyslipidemia and obesity impact COVID-19. Several studies reported associations between low HDL-cholesterol and elevated triglycerides (TG) in serum and severe COVID-19 [43,46,47,48,49,50]. This lipid signature is known as atherogenic dyslipidemia. In addition, many studies support that excess body weight is an important risk factor for severe COVID-19 [51,52]. One study found a linear increase in the risk of severe COVID-19, admission to the hospital and death, starting already at a Body Mass Index (BMI) > 23 kg/m^2^, which is not attributable to excess risks of related diseases. This risk due to elevated BMI was particularly notable in people younger than 40 years of age [53]. However, a study in Costa Rica showed that the host co-morbidities are not specific of a particular clinical profile of patients during pre-vaccination time [54].

**Pre-existing heart failure (HF).** Early in the pandemic, reports exhibited that patients with pre-existing cardiovascular disease had increased mortality from COVID-19 [55]. A study in Italy showed that death was more than double in patients with cardiac disease, including HF, atrial fibrillation and coronary artery disease, compared to patients without cardiac pathologies [56]. HF is a chronic and progressive clinical syndrome in which the myocardium is unable to maintain an adequate cardiac output to meet the body needs for blood and oxygen as a consequence of structural and/or functional cardiac abnormalities [57]. Pre-existing HF was identified as a major factor leading to poorer prognosis in COVID-19 patients [56] and as a determinant for non-cardiovascular organ failure [58]. Approximately half of HF patients hospitalized for COVID-19 died (from all causes) [59], while 10% of the patients hospitalized for COVID-19 had a history of HF with higher levels of troponin (i.e., marker of myocardial injury) [60]. A recent meta-analysis confirmed that pre-existing HF is associated with higher mortality, worse prognosis and higher risk for hospitalization [61]. All this led us to investigate the mechanisms by which pre-existing HF modulate COVID-19.

**Gut dysbiosis.** Many studies show a link between COVID-19 and gut dysbiosis, defined as a reduction in gut microbial diversity and depletion of beneficial bacteria with enrichment of pathogenic microbes. Infection with SARS-CoV-2 directly alters the gut microbiota in mice, hamsters, and non-human primates [62,63,64]. Gut dysbiosis can thus be considered an AO caused by SARS-CoV-2. Studies in humans showed that COVID-19 patients exhibited fecal microbiome alterations at all times of hospitalization compared to controls [65,66,67,68]. In addition to being a consequence, a correlation between dysbiosis and severe COVID-19 has been shown in at least two human studies [65]. Compared to patients with mild symptoms, the gut microbiota of patients with severe COVID-19 showed lower Firmicutes/Bacteroidetes ratio, enrichment of Proteobacteria, and lowered abundance of beneficial butyrate-producing bacteria [69]. In addition, alteration of the gut microbiota was associated with several risk factors for severe COVID-19 [70]. This prompts us to explore how gut dysbiosis acts as a detrimental pre-condition in COVID-19 patients [64].

### 2.3. Lifestyle Factors

**Diet.** Dietary patterns, such as the “Mediterranean” or the “Western” diet, as well as certain foods, are proposed to impact COVID-19 prognosis and account—at least partially—for regional differences in death rates [71,72,73]. Epidemiological analyses highlighted a role for diet in disease prognosis in some cases. Severity was negatively associated with habitual intake of legumes, grains, breads, and cereals [73]. Two studies have associated a “Mediterranean” diet with lower risk of infection and severity [74,75]. Another study found an association between higher frequency of consumption of plant-based foods and lower risk of COVID-19, after accounting for social determinants of health, pre-existing conditions, and measures to reduce viral transmission [76]. However, another similar study with fewer participants found no association [76]. A growing awareness of the impact of diet on health requires an investigation of the effects of dietary intake—both nutrients and types of foods—on COVID-19 infection and prognosis. Here, we will explore the potential direct impacts of foods eaten regularly prior to infection but will not discuss the treatment of COVID-19 patients with food-derived compounds, or the effects of diet-related diseases on COVID-19.

**Vitamin D deficiency.** Vitamin D deficiency affects around half of the population worldwide [77,78]. Vitamin D deficiency prior to infection was shown to increase the risk of COVID-19 severity and mortality in most studies [79,80]. However, some data do not support a correlation between vitamin D deficiency and severity [81]. Conflicting results might be related to the complex interactions of vitamin D with the status of vitamins A and K, calcium, potassium and phosphorus [82]. Furthermore, some studies also suggest a correlation between supplementation of high levels of vitamin D and reduced hospital mortality in COVID-19 patients [83] or with reduced severity, but not mortality [84]. However, a recent Cochrane review found that strength of evidence about the beneficial effect of vitamin D supplementation in COVID-19 patients is low [85]. More research is needed to fully capture the impact of vitamin D on COVID-19. As low vitamin D status prior to infection appears as an impactful MF [79], it prompted us to investigate how a vitamin D deficiency interferes with COVID-19 underlying mechanisms and locate the current knowledge gaps and inconsistencies. 

### 2.4. Environmental Factors

**Air pollution.** Air pollution covers a range of substances released into the atmosphere due to human activities, with some of the most prevalent and researched compounds including nitrogen oxides (NOx), ozone (O3), and fine particulate matter (PM2.5). The aggravating effects of these compounds on lung diseases are well-established [86], and the number of papers speculating on a connection between air pollution and COVID-19 spread and mortality is growing [87,88,89,90]. At the same time, the direct effects of air pollution on the transmission of the virus are debated [91]. This association is often confounded with high population density and other factors known to affect the levels of transmission. While the positive relationship between air pollution and the transmission of the virus is still largely speculative, evidence points to air pollution as a MF increasing COVID-19 severity and mortality [88,90,92,93,94]. The AOP approach is a relevant and effective framework for investigating the mechanistic aspects of the correlations discussed above.

**Chemical exposure, exemplified by PFAS.** Humans are continuously exposed to a mixture of chemicals from different sources, such as from food, drinking water, and consumer products. A number of chemicals are known to cause adverse health effects, including impaired immunity, at current exposure levels [95]. Hence, it is reasonable to believe that exposure to certain chemicals may affect the response to COVID-19 [96]. Per- and polyfluoroalkyl substances (PFAS), a group of several thousand widespread synthetic biopersistent and bioaccumulative chemicals sharing similar molecular structure (including Perfluorooctane sulfonic acid (PFOS) and Perfluorooctanoic acid (PFOA)) have been shown to cause negative effects on metabolism, thyroid function and the immune system [97], particularly when exposure occurs prenatally and/or during childhood [97]. 

Few studies have investigated correlations between PFAS exposure and COVID-19 thus far. A population living in a PFAS-contaminated region in Italy exhibited a higher mortality risk for COVID-19 [98]. In two regions heavily polluted by PFAS in China, a significant correlation between the urinary concentration of 12 PFAS and COVID-19 infection risk was reported [99]. Increased plasma-PFAS concentrations were associated with severe COVID-19 in Denmark, and this tendency remained after adjustment [100]. Finally, a study from the highly contaminated area in Sweden estimated that the incidence ratio of COVID-19 in the adult population was 1.19 compared to a reference population, suggesting a potential link between high PFAS exposure and susceptibility to COVID-19 [101]. More epidemiological studies are needed to assess the strength of the association between PFAS exposure and COVID-19 severity. However, PFAS exposure seems a plausible MF due to their immunotoxic properties and will be explored here as an example of how chemical exposure could modulate COVID-19 [102].

### 2.5. Therapeutic Intervention against COVID-19

The fast-spreading and disruptive nature of COVID-19 pandemic highlighted the need for rapid and reliable identification of therapeutic options. In the first phases of the health emergency, several hundreds of medications were trialed, creating a huge amount of data of divergent quality. However, only a small percentage of the proposed drugs showed to be effective against the virus [103]. Understanding the mechanisms underlying the disease is essential to propose adequate therapeutic interventions against COVID-19. Furthermore, drug repositioning rapidly emerged, being a quick and cost-effective solution to screen large libraries of candidate compounds [104]. Especially when the information on the molecular pathogenesis is limited, repurposing existing drugs has also the advantage of already proven safety profiles and known targets [105,106]. Here, we aimed to explore how drugs used to treat COVID-19 influence the same causal steps as the risk factors described above. The drugs from Table 1 recapitulating the clinical trials undertaken will be discussed.

## 3. Factors Modulating COVID-19: A Mechanistic Understanding via the AOP

The mechanisms by which the factors described above influence COVID-19 pathogenesis are only partially known. Elucidating such mechanisms would provide valuable insight into the clinical and epidemiological findings. Within the CIAO project, the initial and central mechanisms of COVID-19 were depicted via linear AOPs [14]. KEs, KERs and linear COVID-19 related AOPs are stored in the AOP-Wiki where they are identified by assigned unique numbers and then fully described in respective pages. Numbers in the text refer to these AOP Wiki pages (Table 2). The modular aspect of AOPs allows for the development of networks combining individual linear AOPs that share at least one KE [6,131]. The various COVID-19 AOPs developed within CIAO have been integrated within such a network offering a unique overview of the mechanisms underlying the disease (published elsewhere). This is particularly interesting for COVID-19 as it shows the interrelation between the seemingly disparate adverse clinical outcomes. A simplified version of this COVID-19 AOP network is considered in this manuscript, focusing on inhalation as the entry route (Figure 3).

Briefly, in the respiratory tract, the spike (S) proteins of SARS-CoV-2 bind to the ACE2 receptor present on macrophages, and airway and alveolar epithelial cells. Upon binding to ACE2, the S proteins need to be activated through proteolytic cleavage to allow viral entry into the cell via membrane fusion or need endocytosis to enter (KER2056). When inside the cells, the genome of SARS-CoV-2 is translated, replicated, and viral proteins are transcribed, inducing an antiviral response that must be antagonized for the virions to be assembled and released from the cell (KER2310). The binding of S proteins to ACE2 can also dysregulate the physiological functions of ACE2 (KER2311). ACE2 is a known regulator of the renin-angiotensin system (RAS) regulating blood pressure, wound healing, and inflammation. Those initial KERs (Figure 3) are the starting point of many pathways leading to adverse outcomes in COVID-19. Reduced ACE2 activity due to viral binding was associated with elevated levels of PAI-1 [132], a marker of hypofibrinolysis reported in patients with severe COVID-19 [133]. Decreased fibrinolysis and activation of the bradykinin system induce proinflammatory mediator production (KER2356). In parallel, coronavirus production causes death and injury to the infected cells with the release of damage-associated molecular patterns (DAMP) recognized by the TLR family (KER2303). TLR ligands activate the tissue resident immune cells. Activated local innate immune response includes secretion of soluble proinflammatory mediators which recruit immune cells to the infection sites (KER1703), which further amplify the secretion of cytokines and chemokines. The increase of pro-inflammatory mediators and the recruitment of inflammatory cells are hallmarks of inflammation that can be independently measured [134]. Under normal conditions, inflammation is a protective process that combats infection. However, prolonged inflammatory responses have long been known to play a detrimental role in human diseases. Clinical markers of excessive inflammatory response (KER2354) were associated with severe and fatal COVID-19 [135,136]. Dysfunction of the intestinal barrier prior to infection contributes to exacerbating systemic hyperinflammation through gut translocation of toxins and bacteria into the blood circulation (KER2495). Hyperinflammation contributes to broad tissue damage, acute respiratory distress syndrome, multiple-organ failure and ultimately death [137,138].

The focus of our study is to provide a mechanistic understanding of how MFs intervene with some of the processes described in the COVID-19 AOP network (Figure 3 and Table 2). MFs can impact one or several KERs (Figure 4, blue triangle). Interestingly, KEs or AOs in an AOP might also act as MFs and interfere with KERs in another AOP (Figure 4, blue triangle in yellow rectangle), e.g., co-morbidities are adverse outcomes as well as risk factors. In a similar concept, when considering environmental chemicals, two different scenarios need to be distinguished: (i) co-exposure to a chemical (serum levels) concurrent with SARS-CoV-2 infection, and (ii) pre-exposure to a chemical preceding SARS-CoV-2 infection, affecting the development and functioning of the immune system. 

### 3.1. Key Event Relationships Related to Initial Viral Infection

#### 3.1.1. Binding to ACE2 Leads to Viral Entry and Coronavirus Production (KER2056-KER2310)

Following binding of SARS-CoV-2 S protein to membrane anchored ACE2 (mACE2) receptor (KE1739), the virus enters the cell (KE1738). Within the cell, the virus starts to replicate inducing an antiviral response (KE1901) which must be antagonized for the virions to be produced and released (KE1847). In addition, upon SARS-CoV-2 infection, mACE2 becomes soluble (sACE2) through A Disintegrin and Metalloproteinase 17 (ADAM17) sheddase action [139,140,141,142,143,144]. sACE2 is proposed to confer a protective role against SARS-CoV-2 infection by competing with mACE2. sACE2 “decoy” receptors were suggested as potential therapies against SARS-CoV-2 infection [145,146]. However, pharmacological inhibition of ADAM17 attenuated SARS-CoV-2 infection [143].

**Age.** ACE2 protein expression is increased with aging in several tissues [19], including lungs and particularly in patients requiring mechanical ventilation [147]. During aging, telomere dysfunction activates a DNA damage response leading to higher ACE2 expression. Thus, telomere shortening could contribute to make elderly more susceptible to SARS-CoV-2 infection [148].

**Sex.** ACE2 localizes to the X sex chromosome and displays a sex-dependent expression profile with higher expression in female than in male tissues [149]. In addition, the female sex hormone estradiol increases ACE2 expression in females [150]. Furthermore, it has been reported that estradiol inhibits TMPRSS2, needed to facilitate SARS-CoV-2 entry into the cell [151]. Estrogen therapy has been shown to mitigate endoplasmic reticulum stress induced by SARS-CoV-2 invasion through activation of cellular unfold protein response and regulation of inositol triphosphate (IP3) and phospholipase C [152]. Androgen receptors (ARs) also play a key role in increasing transcription of TMPRSS2 [153]. This may explain the predominance of males to COVID-19 fatality and severity, because males tend to have a higher expression and activation of ARs than females, which is due to the presence of dihydrotestosterone [154,155,156]. Different studies have also illustrated that estradiol increases the expression of ADAM17, leading to high-circulating sACE2 potentially neutralizing SARS-CoV-2 and preventing its binding to mACE2 [157]. Thus, E2 might reduce SARS-CoV-2 infectivity through modulation of cellular ACE2/TMPRSS2/ADAM17 axis expression.

**Genetic factors.** Polymorphisms inducing amino acid residue changes of ACE2 in the binding interface would influence affinity for the viral S protein. Evidence exists that K353 and K31 in hACE2, the main hotspots that form hydrogen bonds with the main chain of N501 and Q493 in receptor-binding motif respectively, play a role in tightly binding to the S protein of SARS-CoV-2 [158]. Around the twenty natural ACE2 variants, three alleles of 17 variants were found to affect the attachment stability [159]. Thus, the ACE2 variants modulating the interaction between the virus and the host have been reported to be rare, consistently with the overall low appearance of ACE2 polymorphisms. In this context, it is key to approach both the ACE2 genotypes and the clinical descriptions of the phenotypes in a population-wide manner, in order to better understand how ACE2 variations are relevant in the susceptibility for SARS-CoV-2 infection [33]. In addition, since ACE2 is X-linked, the rare variants that enhance SARS-CoV-2 binding are expected to increase susceptibility to COVID-19 in males [160]. On the other hand, the X-chromosome inactivation of the female causes a “mosaic pattern”, which might be an advantage for females in terms of reduced viral binding [161]. TMPRSS2 single-nucleotide polymorphisms (SNPs) were associated with a frequent “European haplotype” [162], which not observed in Asians, is suggested to upregulate TMPRSS2 gene expression in an androgen-specific way. Thus, there is a need for in vitro validation studies to assess the involvements of population-specific SNPs of both ACE2 and TMPRSS2 in susceptibility toward SARS-CoV-2 infection. The occurrence of a pandemic is related to the genetics of the infecting agent. In the case of SARS-CoV-2, through genomic surveillance it is possible to track the spread of SARS-CoV-2 lineages and variants, and to monitor changes to its genetic code that can influence viral entry and production. Consequently, genomic surveillance is crucial to understand how mutations occurring on SARS-CoV-2 genome influence and drive the pandemic [163]. For example, a recent study [164] highlights that through genomic surveillance it is possible to trace co-infections by distinct SARS-CoV-2 genotypes, which are expected to have a different impact on factors modulating COVID-19. Genomic surveillance of SARS-CoV-2 is able to reveal tremendous genomic diversity [165], and coupled with language models and machine learning approaches, contributes to predicting the impact of mutations (such as those occurring in the spike protein), and thus can better address challenging aspects, like an estimation of the efficacy of therapeutic treatments [166].

**Dyslipidemia and obesity.** Lipids, as important structural components of cellular and sub-cellular membranes, are crucial in the infection process [167]. Changes in intracellular cholesterol alter cell membrane composition, impacting structures such as lipid rafts, which accommodate many cell-surface receptors [168], including ACE2 and TMPRSS2 [169,170]. In an in vitro study, the depletion of membrane-bound cholesterol in ACE2-expressing cells led to a reduced infectivity of SARS-CoV [169]. In vitro, higher cellular cholesterol increased uptake of SARS-CoV-2 S protein; this effect was decreased with Methyl-beta-cyclodextrin, a compound which extracts cholesterol from cell membranes [171]. Higher membrane cholesterol coincides with higher efficiency of COVID-19 entry. Inversely, COVID-19 patients showed lower serum levels of cholesterol, both HDL and LDL. The modulated efficiency of viral entry can be explained by availability of HDL scavenger receptor B type 1 (SR-B1), a receptor found in pulmonary and many other cells. SR-B1 could facilitate ACE2-dependent entry of SARS-CoV-2 [172]. Potential inverse roles of membrane cholesterol and lipoprotein cholesterol in COVID-19 patients requires further investigation [173]. 

Regarding obesity, ACE2 is highly expressed in adipose tissue, thus excess adiposity may drive more infection [174]. ACE2 protein levels in adipocytes were similar between individuals with or without obesity, but obese patients have more adipose tissue and therefore more ACE2-expressing cells [175]. Some studies have also addressed how SARS-CoV-2 dysregulates lipid metabolism in the host and the effect of such dysregulated lipogenesis on the regulation of ACE2, specifically in obesity [176]. In particular, lung epithelial cells infected with SARS-CoV-2 showed upregulation of genes associated with lipid metabolism [177], including the SOC3 gene. This is of interest as viruses may hijack host lipid metabolism to allow completion of their viral replication cycles. A mouse model of diet-induced obesity showed higher Ace2 expression in the lungs, which negatively correlated with the expression sterol response element binding proteins 1 and 2 (SREBP) genes. Suppression of Srebp1 showed a significant increase in Ace2 expression in the lung. Lipids, including fatty acids, could interact directly with SARS-CoV-2 influencing spike configuration and modifying the affinity for ACE2 and thus its infectivity [178]. Evidence-based literature supports that the dysregulated lipogenesis and the subsequently high ACE2 expression in obese patients might be one mechanism underlying the increased risk for severe complications [176].

**Pre-existing HF.** ACE2 mRNA and protein levels, as well as enzymatic activity, were shown to be upregulated in explanted hearts from patients with end-stage HF, as well as in the HF rat model [179,180,181]. Myocytes, fibroblasts, vascular smooth muscle cells, pericytes [182] and endothelial cells of the coronaries [183] express ACE2, while myocytes in patients suffering from heart disease exhibit higher ACE2 expression [184]. Pericytes—the mural cells lining microvasculature, interacting with endothelial cells notably to maintain microvascular stability—exhibited the strongest ACE2 expression in HF patients [185], rendering these cells involved in the coronary vasculature of the myocardium, more susceptible to infection. Furthermore, SARS-CoV-2 infects and replicates in pericytes, and a decrease in their numbers follows [186]. Patients with pre-existing HF showed increased ACE2 levels in myocytes and pericytes, having thereby higher risk of heart injury [185,187]. In addition, sACE2 levels are higher in HF patients [188,189] and sACE2 activity is increased in HF [190].

In contrast to a protective role of sACE2, it has been proposed that viral binding to circulating sACE2 forms SARS-CoV-2/sACE2 complexes, which might mediate infection of cells in distal tissues [191]; hence, pre-existing HF might disseminate SARS-CoV-2 infection. Interestingly, the increase in sACE2 activity is associated with HF with reduced ejection fraction (HFrEF) but not with HF with preserved ejection fraction (HFpEF), suggesting (i) a rather complex role of HF in regulating ACE2-mediated infection by SARS-CoV-2 [188] and (ii) the potential of sACE2 activity to be used as a biomarker to distinguish between the two HF types. Lastly, it is noteworthy that Khoury et al. provided evidence in a different direction, by showing that ADAM17 and TMPRSS2 [192] expression levels are downregulated in a HF rat model, thus potentially conferring a protective role against infection by SARS-CoV-2 in HF [181]. 

**Gut dysbiosis.** Some evidence shows that gut microbiota influences Ace2 expression in the gut. Colonic Ace2 expression decreased significantly upon microbial colonization in mice and rats [193,194]. Coprobacillus enrichment was associated with severe COVID-19 in patients [67] and was shown to upregulate colonic ACE2 in mice [195]. The abundance of Bacteroides species was associated with reduced ACE2 expression in the murine gut [195] and negatively correlated with fecal SARS-CoV-2 load [67,196]. Thus, gut dysbiosis might lead to higher levels of ACE2 in the gut, potentially increasing the ability of SARS-CoV-2 to enter enterocytes. However, while the human gut expresses high levels of ACE2 [197,198,199], and SARS-CoV-2 infection of human enterocytes in vitro is supported by strong evidence [200,201,202], human healthy gut may not be permeable to viral entry due to the gastric pH, antiviral response and/or the protective multi-layers of the intestinal barrier [201,203]. However, individuals with altered mucosal barrier prior to infection might be more vulnerable to gastrointestinal SARS-CoV-2 infection [204]. The colonic mucus barrier is shaped by the composition of the gut microbiota [205]. Thus, gut dysbiosis can contribute to disrupting the mucus barrier, rendering the gut more permissive to SARS-CoV-2. Further research is needed to acquire a comprehensive understanding of the experimental and clinical conditions under which SARS-CoV-2 productively infects enterocytes. 

**Diet.** Several food ingredients impact viral infection [206]. For example, geranium and lemon oils were found to reduce in vitro ACE2 activity and expression, as well as ACE2 and TMPRSS2 mRNA levels [207]. Several molecular modelling and docking studies indicate the potential for compounds found in garlic [208], turmeric (curcumin) [209], thyme and oregano (carvacrol) [210], green tea [211] and other plant foods (quercetin) [212] to inhibit binding of SARS-CoV-2. Pelargonidin, found in red and black berries, was shown to dose-dependently block SARS-CoV-2 binding to ACE2, reduce SARS-CoV-2 replication in vitro and reduce ACE2 expression [213]. Another study supports a role for quercetin and related compounds to inhibit recombinant human ACE2 activity [214] at physiologically relevant concentrations in vitro. Finally, vitamin B12 may inhibit 3CLpro. This vitamin is found in many foods, including seaweed, which also contains other compounds acting on other events in the COVID-19 pathway [215]. In a human crossover study, 30-day supplementation with resveratrol decreased ACE2 in adipose tissue [216], potentially attenuating an increased risk for infection and viral replication in humans with obesity. In vitro, resveratrol inhibited the replication of SARS-CoV-2 [217]. Further evidence comes from SARS-CoV. Many plant-derived terpenoids and curcumin inhibited viral replication in vitro [218]. 

**Vitamin D deficiency.** Vitamin D administration enhanced mRNA expression of VDR and ACE2 in a rat model of acute lung injury [219]. In particular, vitamin D upregulates the sACE2 form [220]. Thus, low vitamin D status may impair the trapping protective mechanism of sACE2 [221]. Furthermore, vitamin D deficiency has been shown to reduce the expression of antimicrobial peptides (β-defensin, cathelicidin), which act against enveloped viruses [222,223]. Decreased sACE2 and cellular viral defense might be some mechanisms explaining how low vitamin D modulate SARS-CoV-2 infectibility. 

**Air pollution.** Exposure to air pollution results in ACE2 overexpression in the respiratory system [224,225,226,227]. Similar effects have been observed on TMPRSS2 expression by particulate matter [226,228], as well as ozone [229]. 

**PFAS.** To date, only one study conducted in mice has reported effects of PFOA on ACE2 and TMPRSS2 (transmembrane protease, serine 2) gene expression. Tmprss2 expression level in mouse lungs increased proportionally at both doses of tested PFOA, whereas the Ace2 mRNA level significantly increased only at the higher dose [230].

**Therapeutic intervention against COVID-19.** Casirivimab, Asirivimab, Imdevimab and Sotrivimab are monoclonal antibodies designed to recognise and attach to two different sites of the Receptor-Binding Domain (RBD) of the S protein of SARS-CoV-2, blocking the virus to enter cells [119,231,232]. Heparin interacts directly with viral particles and has been shown to bind to the SARS-CoV-2 S1 Spike RBD, causing significant protein architecture alteration, impacting infectivity [233,234]. Remdesivir, which is a prodrug of adenosine analogue, binds to the viral RNA-dependent RNA polymerase (RdRp) and inhibits viral replication inside cells through premature termination of RNA transcription [235,236,237,238]. Molnupiravir is an isopropyl ester prodrug, which is cleaved in plasma by host esterases to an active nucleoside analog b-D-N4-hydroxycytidine (NHC) [239]. After entering host cells, it is intracellularly transformed into its active form, β-D-N4-hydroxycytidine-triphosphate (NHC triphosphate) [240,241]. This then targets the RdRp, which is virally encoded and competitively inhibits the cytidine and uridine triphosphates and incorporates Molnupiravir instead. The RdRp (RNA-dependent RNA polymerase) enzyme of SARS-CoV-2 uses the NHC triphosphate as a substrate instead of the cytidine and uridine triphosphates and then incorporates either A or G in the RdRp active centres, forming stable complexes, thus escaping proof reading by the synthesis of a mutated RNA [242,243]. As a result, the virus can no longer reproduce. This mechanism of action (the accumulation of mutations) is referred to as viral error catastrophe [240].

Nirmatrelvir (formerly PF-07321332, Paxlovid™) is an orally bioavailable 3C-like protease (3CL PRO) inhibitor that is the subject of phase 1 clinical trial NCT04756531 and the phase 2/3 clinical trials (NCT04960202 and NCT05011513, NCT04756531, NCT04960202 and NCT05011513). A 3CLpro antagonist will be highly specific to SARS-CoV-2 and will have minimal side effects because 3Clpro shares no homology with human proteases [244,245]. The SARS-CoV-2 genome encodes two polyproteins (pp1a and pp1ab) and four structural proteins [246,247]. The polyproteins are cleaved by the critical SARS-CoV-2 main protease (Mpro, also referred to as 3CL protease) at eleven different sites to yield shorter, non-structural proteins [248,249]. Without the activity of the 3CL PRO, non-structural proteins cannot be released to perform their functions, inhibiting viral replication [250,251,252].

#### 3.1.2. S-Protein Binding to ACE2 Induces ACE2 Dysregulation (KER2311)

Besides viral replication, binding of the viral S proteins to ACE2 can dysregulate ACE2 (KE1854), impeding the physiological functions of the receptor. Notably, mACE2 converts Ang II into Ang (1–7) and Ang I into Ang (1–9). Ang (1–7) and Ang (1–9) bind to their corresponding receptors, MAS1 and AT2R, conferring beneficial anti-inflammatory, anti-fibrotic, anti-thrombotic and antioxidant effects [253]. Overexpression of ACE2 increases virus replication in the lung [254]. On the other hand, occupation of ACE2 receptors by the virus leads to down-regulation of ACE2 protective effects due to lower levels of Ang (1–7) and higher levels of Ang II causing vasoconstriction, hypertension, and induction of pro-inflammatory cascades via AT1R [255].

**Age.** Reduced ACE2 activity induced by viral invasion may be especially detrimental in older adults with baseline ACE2 deficiency, such as older adults with age-associated HF or gut dysbiosis [256,257]. Higher ACE2 expression leads to higher predisposition to be infected but regarding severity, reduction in ACE2 activity with aging might predispose older individuals to severe COVID-19 [256]. A study showed that ACE2 activity is lower in older women, while the same does not occur in males [258], indicating that ACE2 expression regulation might be dependent on both age and sex. In addition, sACE2 expression and associated increased ADAM17 sheddase activity, has been shown to increase with age, with a higher rate of increase in males [140,259]. The exacerbated ADAM17-mediated ACE2, TNF-α, and IL-6R secretion emerges as a possible mechanism for the acute inflammatory immune response in older men.

**Sex.** As previously mentioned, ACE2 localizes to the X sex chromosome, displaying higher expression in female than in male tissues [149], contributing to explain why women have milder disease progression. Lower levels of ACE2 in SARS-CoV-2 patients has been associated with higher rates of severe outcomes [260]. In particular, ACE2 is involved in the protection of acute lung injury [261], as reduction in ACE2 levels after infection has been associated with severe lung injury [262]. Because females have higher ACE2 levels, presumably more ACE2 remains available after viral entry and impairs severe lung and cardiac manifestation.

**Genetics factors.** Currently, many studies focus on the impact of ACE2 SNPs that alter its expression level. However, SNPs that facilitate binding to S protein have not been inspected in a systematic and genome-wide manner. Many authors have postulated that SNPs in the ACE2 gene (Xp22.2) could affect the binding affinity of SARS-CoV-2 [263]. Altered binding between ACE2 and the S protein is expected to affect the RAS cascade, but no conclusive evidence has been identified so far.

**Pre-existing HF.** The dysregulation of ACE2 and of the RAS system is a characteristic of several cardiovascular pathologies having detrimental inflammatory effects, both locally (in the heart) and systematically [264]. Recently, evidence showed that the S protein itself has profound effects on the normal functioning of the cardiac pericytes also by non-infective mechanisms, e.g., by stimulating the pericyte-mediated release of pro-inflammatory factors that can lead to endothelial cell death [265]

**Gut dysbiosis.** There is a lack of complete comprehension of the modulation of intestinal ACE2 by the microorganisms inhabiting the gut and how this impacts the local and systemic RAS. In addition, ACE2 in the gut has a RAS-independent role in regulating dietary amino acid (AA) transport, such as tryptophan [266,267]. Gut bacterial species influence the free AA distribution in the GI tract, as evidenced in germ-free versus conventionalized mice [268]. Thus, alteration of the gut microbiota might impact intestinal AA metabolism while ACE2 is involved in AA dietary transport. This could potentially contribute to worsening COVID-19 outcomes but needs to be further explored.

**Diet.** Several dietary compounds impact the ACE axis. Many proteins found in seaweed have ACE-inhibiting properties and are thought to shift the balance of RAS towards the less inflammatory ACE2/Ang (1-7)/MAS axis [215]. Resveratrol, a stilbene compound found in several plant foods, appears to be able to promote this pathway as well, as it was found in multiple in vitro and in vivo studies to decrease the expression of angiotensinogen, ACE, and AT1R, and increase the expression of the AT2R and Mas receptor [206,216].

**Vitamin D deficiency.** ACE2 is expressed in the human vascular endothelium and the respiratory epithelium [269]. VDR is also highly expressed in the lung tissue [270]. The effect of vitamin D and VDR on RAS occurs via both induction of ACE2/Ang (1-7) and the vasoactive Mas Receptor axis activity and inhibition of renin and of the ACE/Ang II/AT1R axis, thereby increasing expression and concentration of ACE2, MasR and Ang (1–7) [271]. Thus, vitamin D and VDR exert a vasorelaxant, anti-hypertensive modulation of the axis. Supportive evidence is provided by a VDR agonist, calcitriol, that down-regulated RAS activation in a rat model of acute lung injury [270]. The association of low vitamin D status with overactivation of RAS regulated by ACE2 has been observed also in non-infectious diseases [272]. Therefore, low vitamin D status, through a reduced VDR ligand, supports ACE2 dysregulation and ACE2/ACE imbalance, directly impacting the endothelium of lung vessels [273].

**PFAS.** One study showed that PFOA upregulates ACE2 expression in lungs [230]. However, no studies have been carried out on the downstream effect of PFAS on the ACE axis thus far.

### 3.2. Key Event Relationships Related to Central Inflammatory Processes

#### 3.2.1. Decreased Fibrinolysis Increases Secretion of Proinflammatory Mediators (KER2356)

In COVID-19, the downregulation of ACE2 activity was associated with markers of hypofibrinolysis [274]. Fibrinolysis is an essential physiological process resulting in the enzymatic breakdown of intravascular fibrin in blood clots, preventing excessive fibrin deposition by facilitating degradation of (micro)thrombi in any affected organ. Hallmarks of a decreased fibrinolysis include elevated levels of TAFI and PAI-1 inhibitors, a dysregulated uPA/uPAR system, increased fibrinogen, and high levels of CRP (KE1866). Hypofibrinolysis leads to an increase in pro-inflammatory mediators, such as interleukins (IL-2, IL-6) and TNF (KE1496), through activation of pathways such as NF-κB and VEGF co-receptors (KER2356).

**Age.** During the aging process, alterations of coagulation and fibrinolysis have been evidenced. Hypercoagulability with higher plasma concentrations of fibrinogen and factor VIII seems to be the basis of the increased thrombotic tendency occurring with age [275]. Hemostatic changes during aging have been described associated to plasma concentrations of some coagulation factors, such as fibrinogen, factor V, factor VII, factor VIII, factor IX, high molecular weight kininogen and prekallikrein increase in healthy humans in parallel with the physiological processes of aging. Fibrinogen levels increase in response to IL-6, which itself is strongly correlated with aging. Regarding anticoagulant proteins being modulated during aging, heparin co-factor II levels showed an age-related decrease, independently of sex [276]. The fibrinolytic system is also affected in aging and has previously been described as a systemic state of ‘‘thrombotic preparedness’’ with an acquired thrombophilia, characterized by heightened inflammation and impaired fibrinolytic capability [277]. To date, the implication of PAI-1 has been demonstrated in the process of cellular senescence. A null mutation in the *PAI-1* gene was reported to increase aging in humans [278]. Increased PAI-1 production contributes to the multi-morbidity of aging. Both chronological and stress-induced accelerated aging are associated with cellular senescence and accompanied by marked increases in PAI-1 expression in tissues [279]. Furthermore, PAI-1 governs cellular senescence by regulating the extracellular proteolysis of the senescence-associated secretory phenotype (SASP). It has also been demonstrated that miR-146a negatively modulates PAI-1 in senescent cells, preventing an excessive increase in the production of inflammatory mediators and limiting some of the potentially deleterious effects of the SASP [280]. For this reason, PAI-1 is not only a key mediator of cellular senescence and aging but also of aging-related pathologies [279].

**Genetic factors.** The blood group influences thrombogenesis. Factor-VIII von Willebrand factor is lower in people with group 0 and higher blood levels of Factor VIII are associated with higher thrombotic risk [281]. Emerging evidence indicates that COVID-19 patients are at a high risk of developing coagulopathy and thrombosis, conditions that elevate levels of D-dimer [282]. It is believed that homocysteine, an amino acid that plays a crucial role in coagulation, may also contribute to these conditions. At present, multiple genes are implicated in the development of these disorders. For example, SNPs in FGG, FGA, and F5 mediate increases in D-dimer and SNPs in ABO, CBS, CPS1 and MTHFR mediate differences in homocysteine levels, and SNPs in TDAG8 associate with heparin-induced thrombocytopenia. The gene–gene interaction network revealed three clusters that each contained hallmark genes for D-dimer/fibrinogen levels, homocysteine levels, and arterial/venous thromboembolism with F2 and F5 acting as connecting nodes [283].

**Dyslipidemia and obesity.** Lipoproteins play an integral role in hemostasis and thrombosis. Apolipoprotein A1 (ApoA1), a component of HDL, is ubiquitously antithrombotic [284]. Morelli et al. observed significantly increased odds for venous thrombosis with lower ApoA1 and ApoB levels in a large case-control study [285]. ApoA1 prevented thrombosis in mice by upregulating nitric oxide availability [286], while in vitro studies have demonstrated its potential at fostering the anticoagulant protein C pathway [287]. In correlation with other biomarkers, observational studies have shown that low levels of ApoA1 and low levels in ApoB/ApoA1 in COVID-19 patients would potentially be associated with an “anti-fibrinolytic state” [288], as ApoA1 negatively correlated with PAI-1 while ApoB/ApoA1 were positively associated with plasminogen, resulting in reduced fibrinolytic capacity. Thus, the low HDL precondition associated with atherogenic dyslipidemia observed in severe COVID-19 may contribute to coagulopathy via the loss of the antithrombotic effect provided by these lipoproteins.

**Gut dysbiosis.** The role of the gut microbiome in human plasma coagulation and venous thrombosis has gone unexplored to date. The initial proof-of-principle that the intestinal microbiota composition might affect the coagulation system in humans [289] awaits validation. A mechanistic understanding in the context of COVID-19 is also needed.

**Diet.** The evidence for the influence of specific dietary components on fibrinolysis is minimal. A few studies in human populations found indications that plant-focused or plant-based diets improve fibrinolysis markers, including shorter ELT (euglobulin lysis test, an indicator of higher fibrinolytic activity), increased fibrinolytic activity, increased EFA (euglobulin fibrinolytic activity), and decreased PAI-1 [290,291]. Other studies found associations between high meat intake and PAI-1 and PAI-1ag levels, indicating lower fibrinolytic activity [292,293]. However, some studies found no association between dietary patterns or dietary components and fibrinolytic activity [294].

**Vitamin D deficiency.** Low vitamin D status increases the risk of endothelial dysfunction with increased intracellular oxidative stress [295]. In endothelial cells, vitamin D regulates the synthesis of the vasodilator nitric oxide (NO) by mediating the activity of the endothelial NO synthase. High production of reactive oxygen species (ROS) increases NO degradation and impairs NO synthesis: impaired NO bioavailability is an early event toward the development of vascular damage. In this process, vitamin D acts as a protective agent against oxidative stress, by counteracting ROS production and enhancing the activity of anti-oxidative enzymes such as superoxide dismutase [295]. The antiphospholipid syndrome, a human autoimmune disease with thrombotic manifestations associated with low vitamin D serum levels, provides supportive evidence of the prothrombotic effect of vitamin D deficiency [296].

**PFAS.** PFOS, the most studied PFAS, activates NF-κB and significantly induces the production of TNF-α and IL-6 in Kupffer cells [297], in HAPI cells [298] and in microglial cells [299], as well as in the liver of zebrafish, indicating that PFOS facilitate pro-inflammatory cytokines secretion via NF-κB [300].

**Therapeutic intervention against COVID-19.** Heparin enhances the anticoagulant property of anti-thrombin, prevents fibrin formation and inhibits thrombin-induced activation of platelets and other coagulation factors [233,301].

#### 3.2.2. TLR Dysregulation Increases Secretion of Proinflammatory Mediators (KER2303)

TLRs are a family of transmembrane receptors at the forefront of directing innate and adaptive immune responses against invading bacteria, fungi, viruses and parasites [302,303,304,305]. The engagement of TLR with Pathogen-Associated Molecular Patterns (PAMPs) and host-derived damage-associated molecular patterns (DAMPs) induces conformational changes of TLRs that allow recruitment of adaptor proteins such as MyD88, TIRAP, TRIF, and TRAM to control intracellular signaling pathways, including ERK, p38 and NF-κB, driving the synthesis and secretion of appropriate cytokines and chemokines [306]. In silico analysis depicts that the mRNA of NSP10, S2, and E proteins of SARS-CoV-2 are possible virus-associated molecular patterns that bind to TLR3, TLR9, and TLR7, respectively, and trigger downstream inflammatory cascades [307]. Coronaviruses also contain high numbers of CpG signaling motifs that activate the production of inflammatory mediators, including IL8 and IL17 via TLR9 [308]. 

SARS-CoV-2 has been shown to present CpG “hotspots” in E and ORF10 regions [309]. In addition, SARS-CoV-2 spike protein has been found to bind with surface TLRs, including TLR1, TLR4, and TLR6, of which TLR4 showed the strongest protein-protein interactions [310]. Studies showed that TLR2 plays a critical role in sensing envelope proteins of SARS-CoV-2 upstream of the adaptor MyD88 [311,312]. In vivo infection models further showed that TLR2 mediated inflammation plays a driving role in SARS-CoV-2 induced pathogenesis [311]. In addition to these viral PAMPs, host-derived DAMPs and several MFs are also suggested to contribute to TLR dysregulation [38,313]. The serum of COVID-19 patients present high levels of mitonchondrial DNA (mtDNA), which contributes to the secretion of proinflammatory mediators via TLR9 activation [314]. A positive correlation between cell-free DNA and COVID-19 severity was shown [315]. The increased circulating free DNA (cfDNA) in COVID-19 patients generated excessive mitochondrial ROS (mtROS) in cells in a dose-dependent manner, which was inhibited by a TLR9-specific antagonist.

**Age.** In older people, there are studies indicating both an increase and decrease in TLR expression and signalling [316,317,318]. Renshaw et al. [319] showed a decline in TLR expression and function in aged mice, explanatory for increased susceptibility to infections and poor adaptive immune responses in aging. On the other hand, Olivieri et al. [320] reported that the effect of age on signalling events downstream of TLRs is greater than the effect of age on TLR levels. They suggested that inflammaging can be triggered by an impairment of miRNAs/TLR signalling interaction (in endothelial and immune system cells), leading to activation of immune cells over time. Inflammaging is a higher basal inflammatory state in older subjects, which is a major driving force of frailty and common severe age-related diseases. Other complex age-dependent TLRs signalling mechanism include the decreased ability of aged macrophages to fight pathogens, the accumulation of senescent cells in aged subjects, and the increased release of endogenous TLRs ligands from senescence cells.

**Sex**. Male and female subjects both express functionally active TLRs, but sex differences have been reported. TLR3, TLR4 and TLR7 are coded by the X chromosome. Thus, certain TLR gain and/or loss of function polymorphisms have higher clinical prevalence in men. Particularly, TLR7 localizes to an area of the X chromosome known to escape X-chromosome inactivation [321]. TLR7 is more highly expressed at the protein level by female immune cells than by male ones [321]. A study showed that specific TLR7 loss of function variants lead to poor outcome of SARS-CoV-2, which could be explained via the role of TLR7 in responding to SARS-CoV-2 mRNA recognition by inducing the production of the antiviral cytokine interferon-α (IFN-α) [35]. An X-linked recessive TLR7 deficiency was found present in approximately 1% of men under 60 years old with life-threatening COVID-19) [35]. Plasmacytoid dendritic cells of men with these TLR7 variants produce less IFN-α ex vivo, which could explain their poor defence against SARS-CoV-2 [35]. Sex-specific associations between TLR polymorphisms and poor lung function have been reported [322]. Examples include gain of function polymorphisms of rs187084 in the TLR9 gene displaying significantly lower lung function in male swine operators than those with wild type. Additionally, the gain of function polymorphisms TLR9-1237T/C, rs5743836 is a risk for severe sepsis in pediatric critical care patients, with males having a higher risk and a more pronounced allele frequency of TLR9-1237T/C than females [323]. The sex hormone, testosterone, can reduce TLR4 expression and sensitivity, which is proposed to explain in part the less optimal defence during infection in male compared to female [324].

**Genetic factors.** It is well-documented that the TLR expression is determined by genetic variation within the TLR genes [325]. In addition to the already mentioned sex differences in certain TLR polymorphisms, TLR genes also exhibit a distinct population distribution pattern and are the target of selection pressure. Ethnicity disparity in COVID-19 mortality rates were suggested to be explained in part by elevated gene expression of TLR7 and TLR9. In addition, allelic variation in the TLR adaptor protein, Ticam2, influences susceptibility to SARS-CoV infection in mice as Ticam2^−/−^ mice had high susceptibility to SARS-CoV-2 infection [37].

**Obesity.** Obesity influences TLR9 expression, which is higher in visceral compared to subcutaneous adipose tissue depots in mice and obese patients [326,327]. Obesity-induced cell-free DNA fragments released from adipocytes stimulate chronic adipose tissue inflammation and insulin resistance via TLR9 activation [328].

**Pre-existing HF**. TLRs are expressed in the myocardium, with TLR4 being the most abundantly expressed, and TLR2 and TLR3 being present to a lesser extent [329]. TLR4 is upregulated in failing hearts [323,330,331]. The higher expression of TLR4 in HF patients could predispose them towards pro-inflammatory responses. Evidence shows that the S proteins of SARS-CoV-2 can bind to TLR4 directly or activate it via DAMP- and PAMP-mediated pathways, and thus, induce pro-inflammatory mediators, such as IL-1β, IL-6 and TNF-α [332]. The mediator of this TLR-induced activation appears to be NF-κB [333], an essential transcription factor involved in various cardiovascular pathologies [334]. In addition, following infection by SARS-CoV-2 of adult rat cardiac tissue resident macrophage-derived fibrocytes, TLR4 was further activated with a dual effect: it caused the upregulation of ACE2 and induced a pro-inflammatory M1 polarization of macrophages [335], which can further enhance the pro-inflammatory factors secretion. However, the involvement of pre-existing HF in the modulation of COVID-19 via TLRs is still not fully elucidated but deserves further investigation.

**Gut dysbiosis.** TLRs may be considered an interface among the intestinal epithelial barrier, microbiota, and immune system. Studies have revealed that the membrane (M) protein of SARS-CoV can activate IFN-β and NF-κB signaling through a non-canonical TLR-related signaling pathway independent of TRAF3 [336,337,338] and that commensal microbiota regulates the systemic TLR-driven immune response in host myeloid progenitors by priming JAK signaling, which underlines its role in the cytokine release syndrome. However, the causal effect of the microbiome on TLR dysregulation in most human disorders, including COVID-19, remains to be proven.

**Diet.** In mice, expression of TLR2 and TLR4 in circulating macrophages is upregulated by circulating free fatty acids, which are increased with consumption of high fat diets [339,340]. Circulating free fatty acids also activate the NF-κB signaling directly or by activating cellular surface TLR in the hypothalamus, in mice and primary human myotube and adipose cells, leading to increased expression of some pro-inflammatory mediators [341,342]. A high-fat diet was also shown to modulate inflammation via TLR9, as mice lacking TLR9 or receiving a TLR7/9 antagonist had reduced upregulation of specific pro-inflammatory cytokines compared to controls upon high fat diet [343].

**Vitamin D deficiency**. Vitamin D status modulates cytokine production, at least partly through the differential modulation of TLRs. Vitamin D3 down-regulates TLR9 in human monocytes but not TLR3, which resulted in less secretion of IL-6 in response to TLR9 challenge [344]. High-dose oral supplementation of vitamin D3 (4000 IU/day) in human decreased TLR9 protein levels and mRNA expression of TLR3, TLR7, and TLR9 [345].

**PFAS.** PFOA exposure reduces TLR2 and Myd88 expression in zebrafish and induces a dose-dependent increase in IFN and B-cell-activating factor (BAFF) mRNA levels [346]. The TLR/MyD88/NF-kB pathway could be a mechanism through which PFOA interferes with BAFF and IFN expression [347]. In addition, increasing TLR2 expression in zebrafish exposed to PFOA showed a linear correlation with increased levels of MyD88, IL-1β, and IL-21 mRNA levels, indicating that the TRL/MyD88/NF-κB pathway mediates pro-inflammatory cytokine release in zebrafish [348]. Further research would be needed to evaluate if this pathway is relevant to humans.

#### 3.2.3. Excessive Secretion of Proinflammatory Mediators and Accumulated Recruitment of Inflammatory Cells Lead to Hyperinflammation (KER1703-KER2354)

Different types of pro-inflammatory mediators are secreted during innate or adaptive immune response. Interleukins (IL) IL-1, IL-4, IL-5, IL-6, TNF-α and IFN-γ are among the more commonly measured pro-inflammatory mediators (KE1496) and recruit inflammatory cells, such as macrophages and leukocytes, to the site of infection (KE1497). Markedly high levels of notably IL-6, IL-10 and TNF-α have been reported in patients with severe COVID-19 [349]. Patients also show immunological alterations of the cellular compartment with decreased total lymphocyte counts, T-cell exhaustion and defective lymphocyte responses [350,351]. Excessive inflammatory response is one of the main causes of severe and fatal COVID-19 [352]. Clinically, the hallmarks of hyperinflammation include high serum levels of C-reactive protein (CRP), reduction or absence of lymphocytes (lymphopenia), high levels of ferritin and D-dimer, increased lactate dehydrogenase and higher neutrophil-to-lymphocyte ratio (KE1868) [353]. High IL-6 serum levels and accumulation of neutrophils were also proposed to be causal and indicative of hyperinflammation [354].

**Age.** During aging, a subclinical chronic inflammatory response develops leading to an immune senescent state, where pathogen protective immune responses are impaired, but the production of inflammatory cytokines, such as IL-6, is increased. This process is called inflammaging. The persistent IL-6 elevation can induce lung tissue inflammation and mortality. The rate of inflammaging is higher in men and accelerated inflammaging is believed to worsen COVID-19 outcomes [355]. The chronic inflammatory status is associated with a dramatic depletion of B lymphocyte-driven acquired immunity. Aging also attenuates the upregulation of co-stimulatory molecules critical for T-cell priming and reduces antiviral IFN production by alveolar macrophages and dendritic cells (DCs) in response to infection with the influenza virus [356]. 

Consistent with this finding, the ability of DCs and macrophages to elicit CD8^+^ T-cell response and proliferation and to release antiviral cytokines is impaired in elderly individuals [357]. In parallel, these subjects are characterized by a reduced activity of plasmacytoid DCs, the main sources of type I IFNs, which underpin the antiviral response and provide the first-line sentinels in immune surveillance, also in the lung [358]. Aged cynomolgus macaques infected with SARS-CoV-2 developed more severe disease than their young adult counterparts and showed a blunted type I IFN response coupled with an elevated production of proinflammatory factors including IL-6 [359,360]. The elevated production of pro-inflammatory factors may ultimately induce local and systemic damage by unleashing a cytokine storm [361]. Therefore, in older men the fine balance between the antiviral response supported by T lymphocytes, B lymphocytes, and plasmacytoid DCs and inflammation is tilted in favor of inflammation [355]. Thus, age-related changes in the immune system affect mediators and cells of both the innate (inflammaging) and adaptive (immunosenescence) immune responses, fueling exaggerated inflammation in the elderly [362].

**Sex.** Males display a higher innate immune response to SARS-CoV-2 than females, which conditions their cytokine profile. Men have higher levels of the innate immune cytokines IL-8 and IL-18 in circulation [363]. In contrast, females produce higher amounts of the antiviral infection cytokine IFN-α. Moreover, elderly men in particular display autoantibodies against IFN-α more frequently [364]. 

Estrogens are critical regulators of gene expression and functions in innate immune cells, including monocytes, macrophages, and dendritic cells, as well as lymphocytes such as T-helper 1/2 (TH1/2) cells, regulatory T-cells (Tregs), and B cells. One of the major forms of estrogen, estradiol, has been shown to dampen the production of excessive innate inflammatory cytokines by monocytes and macrophages [365]. In the presence of progesterone, CD4+ T-helper cells skew from Th-1 to Th-2 in the production of anti-inflammatory cytokines, specifically IL-4 and IL-10 [366]. The cellular types involved in male and female immune responses to SARS-CoV-2 are distinct. Men display higher circulating levels of non-classical monocytes, while immune response in females is enriched with activated T-cells [363]. In lactating women, higher SARS-CoV-2 reactive memory B-cells and antibody titers have been associated with the hormone prolactin [367]. Poor T-cell response to SARS-CoV-2 correlates with worse disease progression in female patients. Higher innate immune activation in men leads to higher plasma levels of the inflammatory cytokines IFN-α [368], IL-8 and IL-18 [363], driving hyperinflammation and more pronounced lymphopenia in males.

**Genetic factors.** The inflammatory response manifested by increased cytokine levels results in inhibition of heme oxygenase (HO-1), with a subsequent loss of cytoprotection. In the 5′-non-coding regions of the HO-1 gene, there are two polymorphic sites, namely the (GT)n dinucleotide and T (−413) A sites, which regulate the transcriptional activity of HO-1. These polymorphisms have been shown to be associated with the occurrence and progression of numerous diseases, including COVID-19 [369]. The timing of the IFN response to SARS-CoV-2 infection can vary with viral load and genetic differences in host response. When the viral load is low, IFN responses are engaged and contribute to viral clearance, resulting in mild infection. When viral load is high and/or genetic factors slow antiviral responses, virus replication can delay the IFN response and cytokine storm can occur before adaptive responses clear the virus, resulting in severe disease including MIS-C [370].

**Dyslipidemia and obesity.** Lipids impact innate and adaptive immune responses. The lipoprotein dyslipidemia associated with COVID-19 severity (high TG and low total, LDL and HDL cholesterol) was inversely correlated with inflammatory biomarkers such as increased levels of serum CRP, IL-6, IL-8, and IL-10 [288,371]. In obesity, immune cells interact with various classes of lipids, which can control the plasticity of macrophages and T lymphocytes. Altered lipid homeostasis is associated with severe COVID-19 outcomes and, at the same time, with chronic inflammation and inflammatory polarization of macrophages and T lymphocytes [372]. Th1 lymphocytes are more prevalent in adipose tissue of obese patients [373]. In the same way, Th1 lymphocytes are elevated in visceral fat [373]. Both macrophages and T lymphocytes interact with lipids that influence their proliferation, differentiation, polarization [374] and transcriptional regulation, which is tightly controlled by SREBP and LXRs, expressed in macrophages and known regulators of cytokine release. Adipose tissue produces many pro-inflammatory adipokines and cytokines, which lead to low-grade inflammation and the recruitment of immune cells which may clarify the connection between obesity and COVID-19 severity [375]. 

**Pre-existing HF.** Dysregulation of RAS due to pre-existing HF can have detrimental inflammatory effects both locally (in the heart) and systematically. The ACE2/Ang (1-7) pathway is associated with the attenuation of a wide range of pro-inflammatory cytokines and chemokines, such as IL-1β, IL-5, IL-6, IL-12, CCL2, TNF-α and MCP-1 [376]. ACE2 downregulation leads to a shift towards the Ang II/AT1R pathway, and a pro-inflammatory response leading to the recruitment of inflammatory cells, such as monocytes and macrophages in the heart [377,378]. In addition, ADAM17 is implicated in a wide range of cardiovascular pathologies [379] and its expression is increased in HF [380,381]. ADAM17 is known as a sheddase of ACE2, but also as the TNF-α converting enzyme (TACE) [380]. According to Palacios et al. [382], increased levels of ADAM17 are correlated not only with mortality, but also with increased circulation of soluble forms of TNF-α and its corresponding receptors (soluble TNFR1/2)—key mediators of the COVID-19-associated cytokine storm [383,384] and the activation of inflammatory cells like macrophages and neutrophils [385]. Thus, pre-existing HF and associated enhanced ADAM17 expression might predispose an organism to enhanced pro-inflammatory cell activation. Hyperinflammation is one of the hallmarks of HF and counteracting the inflammatory response has been for years a target for various experimental therapies [386]. Crucial pro-inflammatory mediators such as TNF-α, IL-1β and IL-6 have been shown to affect endothelial inflammation, leading to the recruitment of monocytes, themselves secreting cytokines, thus contributing to the cytokine storm [387].

**Gut dysbiosis.** The gut microbiota is increasingly acknowledged to play a central role in human health and disease, notably by shaping the immune response. In a COVID-19 cohort, the depletion of several bacterial species (*B. adolescentis*, *E. rectale* and *F. prausnitzii*, known to play immunomodulatory roles in the human GI system) was linked to increased plasma concentrations of TNF-α, CXCL10, CCL2 and IL-10 [65]. Conversely, two species enriched in the COVID-19 cohort, *B. dorei* and *Akkermansia muciniphila,* were positively correlated with IL-1β, IL-6 and CXCL8. Using a machine learning model [388], it was reported that the disruption of gut microbiota significantly correlated with pro-inflammatory cytokines and may predispose normal individuals to severe COVID-19. In addition, some bacteria living in the gut produce short-chain fatty acids (SCFA), recognized as mediators of the intestinal inflammatory response [389]. SCFAs modulate inflammation by regulating immune cell cytokine production such as TNF-α, IL-12, IL-6 [390]. For example, butyrate decreased the LPS-induced TNFα expression in monocytes [390] and activated Treg cells, blocking an excessive inflammatory response [389,391]. Decreases in the abundance of butyrate-producing bacteria and a decline in SCFA were observed in severe COVID-19 [65,67,392,393]. Reduced relative proportion of bacteria producing SCFA was observed in Syrian hamsters infected with SARS-CoV-2, compared to non-infected controls, with a transient decrease in systemic SCFA amounts [63]. However, SCFA supplementation in hamsters during infection had no effect on inflammatory parameters. Targeted analysis of fecal metabolites showed significantly lower fecal concentrations of SCFAs in COVID-19 patients, which correlated with disease severity and increased plasma concentrations of CXCL-10 and CRP [68].

In addition to the reduction of beneficial bacteria, many studies have linked the out-growth of pathogenic bacteria to local and systemic inflammation [394]. The *Prevotella* pathogens, for instance, are associated with augmented mucosal inflammation, including activating TLR2 and Th17-polarizing cytokine production, stimulating epithelial cells to produce IL-8, IL-6, and CCL20, and thus promoting neutrophil recruitment and inflammation [395]. Blooms of opportunistic pathogenic bacterial genera were observed in hospitalized COVID-19 patients, along with translocation of bacteria from the gut into the systemic circulation [64].

**Diet.** A notable body of evidence links dietary elements to pro-inflammatory mediators relevant to COVID-19. High-fat diets have been linked—in multiple studies—to promote an “inflammatory status” in the gut and subsequently other organs, making it more likely that COVID-19 will affect individuals that regularly consume such diets more severely [340]. Hallmarks of this inflammation include breakdown of the intestinal barrier, release of inflammatory cytokines from the vasculature, liver, pancreas, and other organs, release of macrophages, and increase in circulating lipids. Conversely, compounds found in many plant foods may affect COVID-19 prognosis by blocking inflammatory mediators and pathways. Bousquet et al. [396,397] identified bioactive compounds contained in spices and fermented vegetables, including capsaicin, cinnamaldehyde, curcumin, genistein, gingerol, mustard oil, piperine, wasabi, and sulforaphane, that upregulate the signaling of nuclear factor (erythroid-derived 2)-like 2 (Nrf2), a potent endogenous antioxidant which blocks oxidative stress from the AT1R axis, inhibits overproduction of proinflammatory cytokines and chemokines (including IL-6), and limits the activation of NF-ĸB. There is some in vitro evidence that *Lactobacillus*, found in many fermented foods, works through the same mechanism [71]. Finally, naringin, a compound found in citrus fruits, reduced LPS-induced IL-6 expression levels in vitro [398]. Compounds found in foods may be able to affect hyperinflammation. For example, piperine (found in black pepper), inhibits the production of IFN-γ and IL-2 in human peripheral blood mononuclear cells [399] and neutralizes free radicals and ROS [400]. Linoin, a compound found in lemon, has been found to inhibit CD4+ T-cell proliferation by inhibition of NF-κB translocation in human cells [401]. The isoflavone Biochanin-A similarly prevents cell proliferation and LPS-induced inflammatory mediator release in vitro [402]. Increased fiber intake in humans is shown to reduce C-reactive protein [403]. Parthenolide, a potent phenolic compound, inhibited inflammatory mediators in vitro in microglia, monocytes, macrophages, and neutrophils, including IL-6, NF–ĸB, and TNF-α [404,405]. Arachidonic acid, a fatty acid found in numerous foods, enhances the function of γ-amino butyric acid (GABA) receptors, which attenuates severe inflammatory illness in coronavirus-infected mice. This contrasts with other dietary fatty acids, which induce systemic inflammation [406].

**Vitamin D deficiency.** There is a complex interplay between vitamin D and the immune response to viral infections. Low vitamin D status is proposed to induce upregulation of the TNF-α and downstream of NF–ĸB1 signaling pathway, which regulates inflammatory reactions toward viral infection in macrophages [407,408]. Vitamin D was shown as a potent suppressor of IFN-γ-mediated macrophages response, preventing the release of inflammatory cytokines and chemokines [409]. Thus, release of pro-inflammatory cytokines might be exacerbated in COVID-19 patients with vitamin D deficiency [410].

**Air pollution.** Air pollution worsens chronic airway inflammatory diseases [411]. In particular, PM_2.5_ exposure is associated with pulmonary inflammation [412,413]. The direct effect of air pollution on inflammatory status has been long investigated before the current pandemic. Multiple studies have shown that both short and chronic exposures to fine particulate matter increase the levels of circulating IL-1β, IL-6 and TNF-α in the adult general population, as well as in individuals with genetic predispositions [414,415,416]. A substantial proportion of PM_2.5_ is accounted for carbon-based fine particulate matter [417], which is well-researched in the context of inhalation toxicity. Notably, inhalation or aspiration of carbon-based nanomaterials has been shown to induce the release of similar mediators [418,419,420]. In addition to acting as a priming factor that exacerbates the inflammatory phenotype of COVID-19, air pollution also dysregulates the immune cell activity, posing another substantial danger for tissue damage and altering the susceptibility to concomitant bacterial superinfection [421]. Exposure to fine matter particulate can decrease the ability of the immune system to fight pathogens [422,423], especially affecting the normal function of macrophages and alveolar cells, impairing cilia movement, phagocytic activity, and the secretion of inflammatory mediators [424]. Thus, even though the relationship between air pollution and COVID-19 is robust, further studies are needed to clarify the molecular events underlying the phenotype modulation.

It has been hypothesized that air pollution has an indirect effect on COVID-19 lethality by affecting chronic diseases. This is largely due to inflammation induced by exposure to air pollution [425], which exacerbates previous conditions and results in hyperinflammation. Indeed, accumulating evidence shows that especially fine particulate matter increases the inflammatory status at a systemic, organ, tissue and molecular level [412,413,426,427]. The MAPK-STAT3 pathway activated by the ACE2 receptors, which has a central role in the molecular initiation of inflammation, is upregulated in vivo after PM_2.5_ exposure [428]. At the tissue level, pulmonary macrophages are hyper-activated in the lungs of COVID-19 patients, resulting in a detrimental recruitment of cytotoxic effectors that contribute to tissue damage and sustained hyperinflammation [421]. Similarly, air pollution induces an imbalance between cytotoxic and protective immune effectors [429]. At the systemic level, the imbalance between the beneficial and harmful effects of inflammation is also regulated by the hypothalamic–pituitary–adrenal (HPA) axis and the glucocorticoid response [430]. Interestingly, a possible synergistic effect of the viral and pollutant agents has been suggested [431].

**PFAS.** Several in vitro studies in human-derived cells have shown that PFAS can modify the secretion of pro-inflammatory mediators in a dose-dependent manner [432]. PFOS exposure significantly induced IL-1 IL-4, IL-6 and IL-8 in human lymphocytes [433] and reduced chemokines CXCL8 and CXCL10 secretion in human bronchial epithelial cells while increasing of IL-1α release. In addition, both PFOS and PFOA enhanced IL-1β release in response to Poly I:C [434]. However, the immunomodulating properties of different PFAS seem to be complex and a third study in human leukocytes exposed to PFOS, PFOA, PFBS, PFOSA and PFDA reported that all tested compounds decreased PHA-induced release of IL-4, IL-10, and release of IL-6 and IFN-γ was decreased by PFOS, PFOSA, and PFDA [435]. PFOS was a more potent inhibitor of cytokine production than other PFAS in this study, with effects appearing at lower concentrations, and leukocytes obtained from female donors appeared to be more sensitive to the in vitro immunomodulating effects of PFAS, compared to leukocytes from male donors [435]. In rat study exposed to PFOS, increased serum levels of TNF-α and IL-6 were observed [297,436]. Kupffer cells exposed to PFOS showed cell activation, which was mostly inhibited by anti-TNF-α or anti-IL-6 treatment [297]. Moreover, NF-κB inhibitor and JNK inhibitor significantly inhibited the production of IL-6. Collectively, these findings indicate that PFOS triggers an inflammatory response through the NF-κB/TNF-α/IL-6-dependent pathway [297,436]. Consequently, studies support the hypothesis that the immunomodulating effects of PFAS are mediated via NF-κB pathways, likely connected to these chemicals ability to interact with peroxisome proliferator-activated receptors (PPAR) [437,438]. Such modes of action are shared by at least four major representatives among environmentally relevant PFAS: PFOA, PFNA, PFHxS and PFOS [97]. Overall, the evidence supports that PFAS have a complex immunomodulating effect by dysregulating the cytokine signaling network.

**Therapeutic intervention against COVID-19.** Tocilizumab and Sarilumab are anti-IL-6 receptor monoclonal antibodies, which reduce inflammation [439] by attaching to the IL-6 receptor (as IL-6 receptor inhibitors) [440]. Tocilizumab, a biological drug approved for rheumatoid arthritis, is currently being evaluated for its efficacy against the effects of systemic IL-6 elevation (ClinicalTrial.gov accessed on March 2022, NCT04317092, NCT04320615, NCT04306705) [355]. Baricitinib is an immunosuppressant that blocks the action of enzymes known as Janus kinases, which play an important role in inflammatory processes (JAK inhibitor) [441,442,443,444]. Lastly, low molecular weight heparins (LMWHs) have anti-inflammatory effects by blocking pro-inflammatory mediators (TNF-α, IL-6 and LTB4) [445].

#### 3.2.4. Increased Intestinal Permeability Fuels Hyperinflammation (KER2495)

The intestinal barrier constitutes an essential interface between the environment and the internal milieu of the body. Together with the mucosal barrier and the cellular immune system, the intestinal epithelial cell monolayer, and the tight junction (TJ) proteins act simultaneously as a physical barrier against harmful external substances, as well as a selective barrier. Increased intestinal permeability, sign of an impaired barrier function, enhances the translocation of gut bacteria and of bacterial toxins, such as peptidoglycans (PGN) and LPS, from the intestinal lumen into systemic circulation (KE1931) [446]. Increased levels of LPS in the blood (endotoxemia) activates TLRs, leading to the production of numerous pro-inflammatory cytokines and, hence, low-grade systemic inflammation [447]. In ARDS and critically ill patients with sepsis, bacterial translocation is widely documented, and intestinal barrier disruption is considered as an event perpetuating systemic inflammation (KE1868), [353,448,449]. Higher plasma levels of gut permeability markers, PGN and LPS were found in COVID-19 patients, along with abnormal presence of gut bacteria in the blood [65,450,451]. These markers correlated with higher levels of CRP and higher mortality rate [451]. Disrupted intestinal barrier and associated bacterial translocation is proposed to play an additive or synergistic role in the cytokine storm underlying severe COVID-19.

**Age.** Disruption of the intestinal barrier was associated with aging in baboons, along with upregulation of inflammatory cytokines [452]. However, a cross-sectional study in humans assessing gut permeability by validated multi-sugar test and by expression of intestinal barrier-related genes showed no differences between healthy young adults and elderly [453]. Thus, although age-related medication or co-morbidities may impact barrier function, there seems to be currently no indication of impaired intestinal barrier by aging per se in humans.

**Genetic factors.** The epithelial cells of the intestinal barrier express TLRs, which upon recognition of LPS induce epithelial cell proliferation, secretion of mucins, and antimicrobial peptides into the lumen, thereby promoting intestinal barrier function. Polymorphisms of TLR2, TLR4 and TLR9 contributing to individual susceptibility to inflammatory bowel disease, characterized by chronic intestinal inflammation and intestinal barrier disruption had originally attracted attention, but currently no strong association has been observed [454,455,456]. Further research is needed to evaluate the impact of TLR polymorphisms on intestinal barrier in the context of COVID-19.

**Obesity.** In obese mice, gut microbiota was shown to regulate metabolic endotoxemia and associated inflammation through intestinal permeability [457]. In addition, obesity can alter the gut microbiota [458,459,460,461,462,463] and escalate intestinal permeability, enhancing the translocation of bacteria and LPS from the intestine to the blood and adipose tissue, which fuels systemic inflammation.

**Gut dysbiosis**. The gut microbiota ensures the integrity of the intestinal barrier through multiple mechanisms, either by releasing antibacterial molecules and anti-inflammatory SCFAs or by activating essential cell receptors for the immune response [464,465]. The reduction of beneficial butyrate-producing bacteria contributes to increased intestinal permeability, as butyrate facilitates the regeneration of colonocytes [466,467]. Overgrowth of pathobionts, such as *Escherichia coli* or *Salmonella enterica*, disrupts intestinal barrier function, enhancing permeability [468,469,470,471]. Lower levels of butyrate-producers and higher levels of pathogens (including *E. coli* and *S. enterica*) have been observed in COVID-19 patients compared to healthy controls [392,472], and changes in gut microbiota composition correlated with plasma levels of tissue damage markers [65]. Another study associated COVID-19 severity-related gut microbial features (higher abundance of four microbial species and ten virulence genes) with higher levels of inflammation biomarkers and lower levels of immune cells and markers of gut barrier dysfunction in COVID-19 patients [473].

**Diet.** Dietary components or patterns can affect intestinal permeability. A “Western style” diet has been identified as strongly associated with increased intestinal permeability [474]. A recent review postulates that high-fat diets increase intestinal permeability by directly and indirectly disrupting TJs, inducing oxidative stress, and increasing pathogenic intestinal bacteria, which further damage barrier integrity [475] and enhance systemic inflammation. Fibrous foods and foods containing prebiotics boost intestinal barrier function [476].

**Vitamin D deficiency.** Vitamin D deficiency was shown to promote intestinal mucosal barrier dysfunction with higher permeability in infection-induced or TNF-α-treated cells and in in vivo colitis models [477,478]. An association between increased markers of intestinal permeability and vitamin D deficiency has been observed in critically ill subjects from ICU [479].

**Air pollution.** Evidence from animal models suggests that exposure to particulate air pollution may disrupt the intestinal barrier by inducing inflammation, making it a more susceptible site for the entry of pathogens [480], and thus contributing to hyperinflammation.

**PFAS.** Accumulation of PFOA was observed in the gut tissues of orally-treated mice along with lower expression of many TJ genes [481,482]. PFOS was shown to alter gut microbiota, resulting in a decreased bacterial metabolic activity, which in turn altered gene expression damaging intestinal tissue [483]. In mice, chronic exposure to PFOS decreased the expression of the TJ genes in the intestine and reduced the height of the intestinal villi, indicators of altered intestinal barrier [484]. PFOS exacerbated macrophage and neutrophil recruitment to the intestine in both zebrafish and mice and increased epithelial permeability in mice, which led to a systemic expansion of CD4+ T-cells in a neutrophil-dependent manner. This indicates that PFOS can worsen inflammation-induced intestinal damage with the interruption of T-cell homeostasis beyond the gut [326].

## 4. Lessons Learned

### 4.1. Identification of Knowledge Gaps Guiding Further Research

In this study, we investigated how eleven factors impact qualitatively the infectious and inflammatory mechanisms in COVID-19. This approach highlighted current uncertainties, inconsistencies and knowledge gaps guiding further research. An obvious major current limitation is the lack of quantitative data for describing KERs and, thus the quantitative impact of individual factors on KERs cannot be defined. Building disease pathways using the AOP framework helps to set up the scene for quantitative models. 

**Age**. Age is the strongest risk factor for fatal COVID-19. However, not all older subjects were severely ill or died. New concepts emerged on the relationship between immunosenescence/inflammaging and COVID-19. In some cases, the immune response may be adequate and able to face external challenges, but in other cases both immune response and senescence are inadequate to counteract external challenges. This may be due to the role of genetics and the environment but also, on some occasions, the hermetic role of some diseases may lead to aggravate or improve the response of SARS-CoV-2 infection. Thus, there is still the need of understanding immunosenescence/inflammaging process in a more complex systems biology [485].

**Sex.** By investigating the underlying mechanisms, differences between sex can, at least partly, be explained by the influence of hormones, especially oestrogen and testosterone, on both innate and adaptive immune responses [29,156,486], and by sex differences in ACE2 expression, ACE2 and TMPRSS2 regulation [29,156,487,488,489]. Many immune-associated genes and the gene encoding ACE2 are positioned on the X-chromosome and, therefore, females have two copies of these genes; usually one set is silenced, but some genes escape silencing and may offer a protective mechanism for females [490].

The Sex, Gender and COVID-19 Project tracks COVID-19 sex-disaggregated data worldwide and prepares a monthly report for sex-disaggregated such data along the clinical pathway [491,492]. The last available report from November 2021 indicates that 57% of countries provide sex-disaggregated data for cases and/or deaths in the past month, and 41% of countries report on both, cases and deaths [28]. Even though significant progress has been made in recent years, sex-disaggregated data is still scarce in biomedical research in general, and in COVID-19 studies in particular. A recent study collated data showing that among the 2484 registered SARS-CoV-2/COVID-19 trials, only 416 (16.7%) mention sex/gender as recruitment criterion and only 103 (4.1%) allude to sex-disaggregated analysis [493]. To gain a better insight into sex differences, recruitment of an equal number of male and female study participants must be complemented with sex-targeted questionnaires and reporting. For transgender and non-binary people, there are no sex-disaggregated COVID-19 data available, which impedes the identification of specific health care needs of people with diverse gender identities. 

**Genetic factors.** Genetics related to ACE2 receptors and TLRs are critical for virus entry and disease progression. Routine DNA polymorphisms tests able to predict symptomatic COVID-19 would be helpful in distinguishing between subjects at high and low risk, and in estimating disease severity [494]. In addition to ACE2, TMPRSS2 and TLRs, other genetic elements like Apolipoprotein E (APOE) have been reported to increase risk of severe COVID-19 [495,496]. Individuals carrying an allele of CCR5 (CC chemokine receptor 5) with a 32 bp deletion were also more protected against symptomatic COVID-19 [497]. CCR5 is located within proximity of the LZTFL1 (leucine zipper transcription factor-like 1) gene, where a polymorphism was associated with a ∼70% increased risk of hospitalization due to COVID-19 [498]. A gene cluster on chromosome 3 inherited from Neanderthals and carried by ∼65% of the population in South Asia, ∼16% in Europe and almost absent in East Asia was also identified as a risk locus for severe COVID-19 [499]. Another haplotype on chromosome 12, also inherited from Neanderthals, has been associated with ∼22% reduction in relative risk of severe COVID-19. This haplotype is present at substantial frequencies in all regions of the world outside Africa, and it includes the 2′-5′-oligoadenylate synthetase (OAS) 1, 2, and 3 genes, which encode proteins of the 2-5A synthetase family, considered to play a key role in the innate immune response to viral infection [500]. Specific splice variant of OAS1 is likely to be the SNP responsible for the association at this locus, implicating OAS1 as an effector gene influencing COVID-19 severity [501].

**Dyslipidemia and obesity.** Several observational cohorts concordantly agreed that atherogenic dyslipidemia and obesity are associated with a poorer prognosis in COVID-19 patients [502]. However, some studies analyzing the association of total and LDL cholesterol related disorders with COVID-19 severity have shown opposite associations [43]. Some aspects need to be further explored, as there is still data lacking concerning the effect of comorbidities, the impact of medications taken by hypercholesterolemic patients [503] and the specific molecular mechanisms involved in these processes. Evidence shows that lipids play a key role in SARS-CoV-2 entry into cells, due to ACE2 receptor expressed in lipid rafts. However, the role of apolipoproteins and their receptors in the penetration of SARS-CoV-2 into cells requires further research [173]. Another important aspect to be further explored is the role of SR-B1 in penetration of SARS-CoV-2 into the host cell in humans, as recent animal studies demonstrated that an SR-B1 antagonist reduced SARS-CoV-2 infectivity [43,178,504].

**Pre-existing HF.** Despite intensive basic and clinical research, there are still many aspects of HF that are poorly understood. As regards to COVID-19, major gaps in our knowledge are the temporal aspect, the type of HF (acute vs. chronic, HFpEF vs. HFrEF), as well as the way that HF severity can influence the modulation of the prognosis and the course of the disease. In addition, there are very limited studies aiming to mechanistically explain how HF modulates the prognosis of the SARS-CoV-2-infected subject, pericytes are largely understudied as myocardial cell types, and a good deal remains to be understood about their intricate roles in the healthy and damaged heart [505].

**Gut dysbiosis**. A body of evidence indicates that gut dysbiosis contributes to more severe outcomes in COVID-19. The intimately linked role of gut microbiota with local and systemic inflammation is increasingly recognized in COVID-19 and other diseases [506]. However, the mechanisms are still poorly understood. Notably, there is no optimal “healthy” gut microbiota profile since it is unique for each individual. It is still not fully clear which microbiome components contribute to health or disease. Too few microbiome data sets currently exist to enable robust conclusions (around 10,000 publicly accessible samples compared to 30 million human genomes sequenced [507]. Further investigation of the gut microbiome by observational and interventional studies is needed. Elucidating how to maintain or restore a “healthy” gut is relevant not only for COVID-19 but also for numerous diseases, such as chronic, inflammatory, metabolic, and neurological disorders [508].

**Vitamin D deficiency**. Growing evidence supports a role of vitamin D deficiency as a risk factor for severe COVID-19, based mainly on biological plausibility. Vitamin D acts by binding to the specific VDR nuclear receptor. In addition to its well-recognized role in calcium-phosphorus homeostasis, vitamin D regulates the differentiation of several cell types, including immune system cells [509]. However, the interplays between vitamin D and other nutrients in modulating the functional differentiation of immune cells and their ability to respond to infections, including COVID-19, still need to be further investigated [82].

**Diet.** Evidence points to clear impacts of foods and food compounds on COVID-19 prognosis. Mechanistically, it can be shown that compounds found in certain foods can impact the immune system and inflammatory status. A connection between some dietary compounds and SARS-CoV-2 infection is supported by computational, in vitro, and in vivo studies. However, there are several gaps in the body of evidence. Diet is complex and multi-factorial, may change from day to day, and is difficult to measure in human observational studies in a way that could link amounts of specific foods or compounds to COVID-19 prognosis. Consideration of bioavailability and dose is also required and can change depending on characteristics or disease status of the patient. Furthermore, it is likely that combinations of bioactive compounds in foods, rather than compounds in isolation, affect COVID-19 prognosis [510]. Given this, much of the literature evidence is qualitative rather than quantitative, impacting our ability to demonstrate “how much” of certain types of foods affect the KERs discussed here. Differences in metabolism, nutritional needs, and immune system function between humans and other animals mean that we should exercise caution in applying evidence from other animals or animal cells to humans. Additional epidemiological studies assessing food and nutrient intake are warranted, as well as in vitro studies utilizing human cell and tissue models. Indeed, three-dimensional models of the lung, colon, heart and vascular, liver, kidney, brain, and other tissues on microphysiological system platforms are being used to study the mechanisms of SARS-CoV-2 infection and subsequent effects [511,512]; experiments assessing the impacts of nutrients or other compounds contained in foods on COVID-19 disease process in such models would be well-received.

**Air pollution.** Multiple studies have suggested a complex relationship between air pollution and occurrence and severity of COVID-19, either directly or indirectly. The direct role of air pollution in exacerbating inflammation in the lung is well-characterized, as is its ability to prime immune populations by making them more prone to uncontrolled hyperinflammation in response to SARS-CoV-2 infection. The indirect effects on COVID-19 mortality through the role of air pollution as a risk factor for chronic lung and vascular comorbidities is well-established. Since exposure to air pollution was shown to result in ACE2 overexpression in the lung [224,225,226,227] and to negatively affect the protective mechanisms of the respiratory system [513], these mechanisms together gave rise to the “double-hit hypothesis” where their combined effects increase susceptibility to SARS-CoV-2 infection, as well as COVID-19 severity [514]. 

At the same time, controversial findings and emerging data highlight the need for a better understanding of the effects of air pollution exposure on COVID-19 pathogenesis. For instance, it is postulated that air pollution could carry the virus deeper into the lungs and impair the proper functionality of the lung epithelia, reducing the clearance of the virus and contributing to the development of more severe symptoms [515]. Similarly, air pollution has been suggested as a carrier for the viral particles directly affecting the transmission of the virus [516,517,518,519,520]. These ideas, however, have been subjected to heavy critique, requiring thorough investigation [91]. Further research should also be targeted towards understanding the ways in which different pollutants can modulate the innate immune response into distinct directions. Finally, the mechanisms of different constituents of air pollution, and whether mixtures have a synergistic or antagonist effect on the immune system have not yet been clarified to date [521].

**PFAS**. A growing body of evidence indicate that PFAS may act as immunomodulators in humans, particularly when exposure occurs prenatally and/or during childhood. However, the mechanisms by which PFAS may enhance the susceptibility and/or weaken the immune response in COVID-19 are still insufficiently known. Furthermore, a distinction should be made between the effect of the co-exposure to PFAS (serum levels) concurrent with SARS-CoV-2 infection, and of the developmental exposure to PFAS. While PFAS are relatively well-characterized in terms of their mechanisms and effects possibly impacting COVID-19, far less is known for many other environmental chemicals. Feeding the knowledge about PFAS provides better understanding and a blueprint for investigating how environmental chemicals may modulate disease in the human population.

**Therapy against COVID-19.** Better understanding of the COVID-19 mechanisms has allowed an improved treatment over the distinct stages of disease. Continuing to understand the pathogenesis is needed to support the search for new and reliable therapeutics to expand our COVID-19 arsenal. Preclinical data and clinical trial results on antivirals, immunomodulators, cellular and gene therapies, and neutralizing antibodies emerge regularly. As of February 2022, more than 100 compounds have reached late-stage trials, and a handful have emergency-use authorization or are approved [522]. The World Health Organization (WHO) maintains an exhaustive list of COVID-19 trials [523] and the U.S. FDA Coronavirus Treatment Acceleration Program (CTAP), an overview of the types of drugs being studied for the treatment of COVID-19 [524].

### 4.2. Biomarkers Related to MFs as Predictive Tools

Biomarkers are measurable biochemical effects that can be associated with an established or possible disease [525,526]. AOPs can provide mechanistic details to identify KEs that can serve as reliable anchors for biomarkers [527].

**Age**. Biomarkers related to inflammaging or immunosenescence could be predictive for COVID-19 outcomes [528]. Most severe cases demonstrated elevated levels of infection-related biomarkers, inflammatory cytokines and cellular senescence related to aging. Increased serum levels of cell free DNA including mtDNA and ssDNA were also proposed as biomarkers for COVID-19 vulnerability [38,522]. Telomere shortening, a hallmark of aging, causes depletion of certain telomeric sequences in cytoplasmic and cell-free DNA pool, which is being proposed to influence age-related activation of the innate immune system [529]. Circulating mtDNA levels were shown to increase with “inflammaging” [530] and cell-free DNA was shown to map COVID-19 tissue injury and risk of death [530].

**Genetic factors.** More than 100 COVID-19 biomarkers have been identified [531] by developing disease biomarker data models that are continuously being improved. These allow integration and evaluation of biomarkers in patients with comorbidities. Most biomarkers are related to the immune (SAA, TNF-∝ and IP-10) or coagulation (D-dimer, antithrombin and VWF) cascades, suggesting complex vascular pathobiology of the disease [532]. Linking these new biomarkers with the individual host genome mapping exercise will identify new strategies for personalised treatment therapies. 

**Dyslipidemia.** Studies showed that serum low HDL and high TG levels measured before or during hospitalization can be strong predictors of severe COVID-19 and should be considered a sensitive severity biomarker [48,533].

**Pre-existing HF.** A meta-analysis to investigate cardiac biomarkers in COVID-19 revealed several of them (cTnI, cTnT, hs-cTn, hs-cTnI, hs-cTnT, MB, CK-MB) to be directly associated to worse prognosis of COVID-19 [534]. Troponin I level measurement is routinely used to diagnose acute myocardial infarction; however, it can also be elevated in acute HF [535] independently from acute myocardial injury. Higher troponin I levels in the first 24 h post-admission are associated with dramatically enhanced mortality risk [536] and could thus be used as a useful tool in predicting in-hospital mortality. Furthermore, N-terminal pro-B-type natriuretic peptide (NT-proBNP) is released when the myocardium is under increased myocardial wall stress and is a well-described marker of HF. Severe COVID-19 patients with high levels of NT-proBNP had a significantly higher mortality risk in comparison to patients with low levels [537,538], advocating for NT-proBNP as a biomarker to predict the risk of in-hospital death in COVID-19 patients.

**Gut dysbiosis.** Hospitalized patients with severe COVID-19 showed microbial signatures of gut dysbiosis, pointing out gut microbiota diversity as a prognostic biomarker of COVID-19 severity [69], with a limitation, however, due to the considerable variation in microbiota composition across populations. Gut molecular profiles obtained by metaproteomics, metabolomics and lipidomics provided a shortlist of microbial and human biomarker candidates indicative of intestinal SARS-CoV-2 infection [393], which could help monitor infection in general.

**Diet.** Significant effort has been put into the validation of food intake biomarkers to supplement findings of food intake questionnaires during dietary studies. Some of the considerations include metabolism of the components of certain foods, frequency of consumption and duration since last intake, and complexity of diets containing many different compounds. It is possible, however, to assess the regular circulating levels of compounds found in food that might be relevant to COVID-19 prognosis, such as quercetin, lycopene, and lipids [539]. Databases have also become available offering information on the amounts of certain constituents in food, such as polyphenols (Phenol-Explorer database [540], accessed on 16 March 2022). Validated biomarkers together with databases or similar tools may help to improve the specificity of human clinical and epidemiological studies assessing the impact of diet on COVID-19.

**Vitamin D deficiency**. According to EFSA, serum 25(OH)D concentration can be used as a biomarker of vitamin D status at population level. Additional biomarkers are the serum concentrations of free 25(OH)D, 1,25(OH)2D and parathormone. Dietary reference values for vitamin D in the EU population are available [82].

**PFAS**. Since PFAS persist in organisms almost unmetabolized, the compounds themselves represent the biomarkers of exposure for biomonitoring studies on humans. Early biomarkers of effective dose/effect are linked to the knowledge of mechanisms. In case of PFAS, such biomarkers could be related with PPAR- and NF-κB signaling pathways.

### 4.3. Risk Factors Impact on Therapeutic Interventions Effects

Risk factors, such as sex, age and co-morbidities, might impact the effect of therapeutic interventions against COVID-19 described in this study.

Regarding age, Molnupiravir is not authorized for use in children aged < 18 years due to potential effects on bone and cartilage growth [541]. Sarilumab is not approved in children, so if an IL-6 receptor blocker is used in this population, tocilizumab is preferred [542]. Furthermore, sex can influence the pharmacokinetics, pharmacodynamics, and the safety profile of drugs but as mentioned earlier, very few SARS-CoV-2 clinical studies explicitly report a plan to include sex a modifying variable [543,544,545]. 

Potential drug–drug interactions may occur if patients are under medication for an underlying medical condition [546]. Drug interactions occur between Remdesivir and drugs commonly prescribed in dentistry [547]. Ritonavir (in Paxlovid) is an inhibitor, inducer, and substrate of various other drug-metabolizing enzymes and/or drug transporters [548,549,550]. In this way, it may also increase concentrations of certain concomitant medications, thereby increasing the potential for serious drug toxicities. High IL-6 concentration may down-regulate CYP activity. A blockade of IL-6 signaling by IL-6Rα antagonists, such as Sarilumab or Tocilizumab, might reverse the inhibitory effect of IL-6 and restore CYP activity, leading to increased metabolism of drugs that are CYP substrates (Sarilumab (Kevzara) [551], Tocilizumab (Actemra) [552]).

### 4.4. Interplay between Factors Modulating COVID-19

Factors impacting COVID-19 do not only act in isolation, but there is also a complex interplay between them, both at the level of the general population and individuals (Figure 5). Looking at the mechanisms of the modulating effect, these factors share several KERs, thus they act on similar mechanisms. Therefore, the concurrence of several factors should be taken into account for prognostic and preventive measures. As for the latter, healthy diet or lifestyle habits can be encouraged and exposure to environmental stressors can be reduced at individual, local and global levels.

In particular, the interplay between MFs and adverse effects identified as MF themselves—e.g., HF, dyslipidemia and gut dysbiosis being both MFs and AOs (in red in Figure 5)—renders the understanding of this interplay even more challenging. Indeed, age, sex, obesity and lipid disorders are well established factors influencing the course of **HF** [553,554]. There is also a strong correlation of HF-related mortality and morbidity with air pollution [555]. Observational studies highlighted the relationship between exposure to PFAS and cardiovascular disease [556]. Quite recently, a cross-talking mechanism has been established between gut dysbiosis and HF [557] with evidence pointing towards the presence of a heart–gut axis [558], which disturbances in the normal intestinal microbiota, chronic systemic inflammation and HF.

The **gut microbiota** itself is shaped throughout an individual life. It is altered by many factors detailed as MFs here, such as elderliness, obesity, HF, diet, pollution and environmental chemicals [462] in ways that may strongly alter both the susceptibility to SARS-CoV-2 infectibility of the GI tract, as well as the local and systemic inflammatory response [70]. With aging, there is an unbalance between the beneficial and pathogenic gut bacteria, increasing pro-inflammatory mediators potentially contributing to inflammaging [485]. There is also a known mutual impact of the intestinal microbiota in obesity [559]. Consumption of high-fat diets affects the microbiota and increases intestinal inflammation and permeability; therefore, high fat diets may synergize the impacts of gut dysbiosis on COVID-19 prognosis [340]. Recent evidence suggests that the gut microbiome may be directly impacted by and/or alter the effects of PFAS [483,484,560].

Certain dietary patterns, lack of exercise, advanced age, or male sex increase the risk of **dyslipidemia, obesity** and metabolic related comorbidities, all these having an impact on COVID-19. **Vitamin D deficiency** is also a common finding in obesity [561] and is also related to gut dysbiosis, dyslipidemia, diet and age [562]. PFAS have structural homology with fatty acids and may interfere with lipid metabolism [563]. A positive association between exposure to PFAS and obesity was found [564]. Limited in vitro and in silico research suggests that PFAS may also affect vitamin D homeostasis [565]. Exposure to chemicals often differ between age groups and children (including toddlers) have about a twofold higher exposure to many PFAS than adolescents, adults and the elderly [97].

Based on the current knowledge, even if incomplete, it is worthwhile to consider gut microbiota modulation with probiotics, prebiotics or diet as a potential prophylactic approach to mitigate COVID-19 [566]. A few reports cite indirect evidence for the association between probiotics and COVID-19, primarily based on previous coronavirus and other viral infections [567,568]. Using conventional probiotics is not warranted currently but is considered promising, urging a better understanding of SARS-CoV-2 pathogenesis and their mutual effects on gut microbiota is needed. Diet affects nutritional status, obesity and lipid status, and may also be a source of pollutant exposure. Consumption of a diet containing toxic-free foods and consistent with the current healthy eating guidelines might strengthen the immune system [569] and positively impact other conditions such as cardiovascular diseases, lipid disorders, gut dysbiosis or vitamin D deficiency, which are identified as risk factors for COVID-19.

## 5. Conclusions

Substantial variation in COVID-19 outcomes is observed between patients, from no or very few symptoms to severe lung or multiorgan failure to death, calling for identifying factors modulating the outcomes and understanding their mechanisms of action. This is essential for improving public health measures, especially to protect vulnerable populations. This paper illustrates how clinical and epidemiological data, along with mechanistic evidence, helps assess the impact of a wide variety of intrinsic and extrinsic factors on the sequence of events underlying the disease. Applying the AOP framework allows us to draw on existing knowledge about the mechanisms by which some factors modify biologic processes and thus impact the course of a disease. These factors often act upon the same steps but also interact with each other. These complex interactions may increase or attenuate processes that have not yet been investigated in detail. Quantitative data are especially missing, and are needed notably for building predictive models. Elaborating disease pathways provides a framework for such modeling. Furthermore, due to the complexity and fast-evolving feature of the disease, there is a need for an interdisciplinary approach that is fully endorsed by this AOP-aligned methodology.

## Figures and Tables

**Figure 1 jcm-11-04464-f001:**
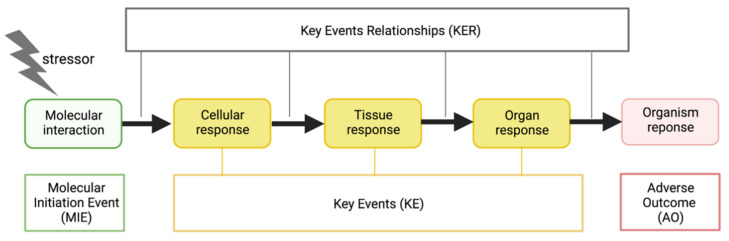
An AOP describes a sequential chain of causally linked events at different levels of biological organization that lead to an adverse health effect. Created with *Biorender.com*.

**Figure 2 jcm-11-04464-f002:**
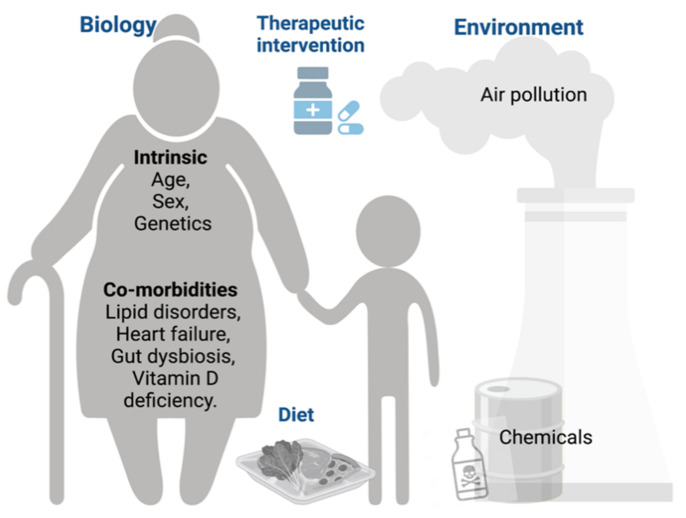
Factors modulating the clinical outcomes of COVID-19 investigated in this study, being representative of different categories: intrinsic (age, sex, and genetic factors), co-morbidities (dyslipidemia, obesity, pre-existing heart failure, and gut dysbiosis), lifestyle-related (vitamin D deficiency and diet). and environmental (air pollution and exposure to chemicals). Created with *Biorender.com*.

**Figure 3 jcm-11-04464-f003:**
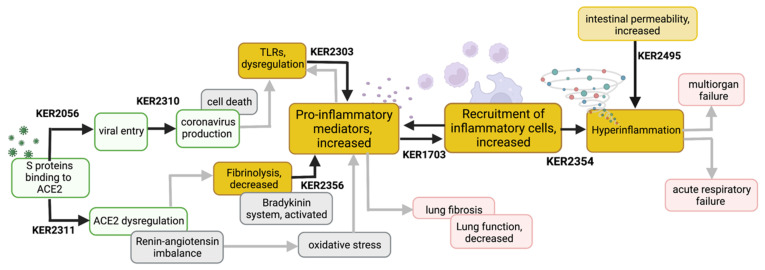
Simplified AOP network depicting COVID-19 pathogenesis, highlighting the biological key steps for evaluation of the mechanisms by which the eleven selected MFs affect the relationships between two KEs, namely the KERs (black arrows). Green boxes: initial events depicting the infectious process and ACE2 dysregulation. Orange boxes: central inflammatory events. Red boxes: AOs. Grey arrows and grey boxes: KERs and KEs identified in COVID-19 but not investigated in detail here. Created with *Biorender.com.*

**Figure 4 jcm-11-04464-f004:**
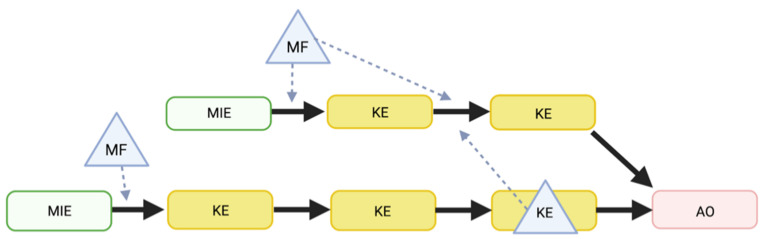
Visualization of MFs within an AOP network. Individual MFs (blue triangle) might have an influence (dotted line) on one or several KERs within an AOP or within a network. In addition, a KE (blue triangle within the rectangle) can act as a MF for KER(s) in other AOP(s). Diagram adapted from US EPA. Created with *Biorender.com.*

**Figure 5 jcm-11-04464-f005:**
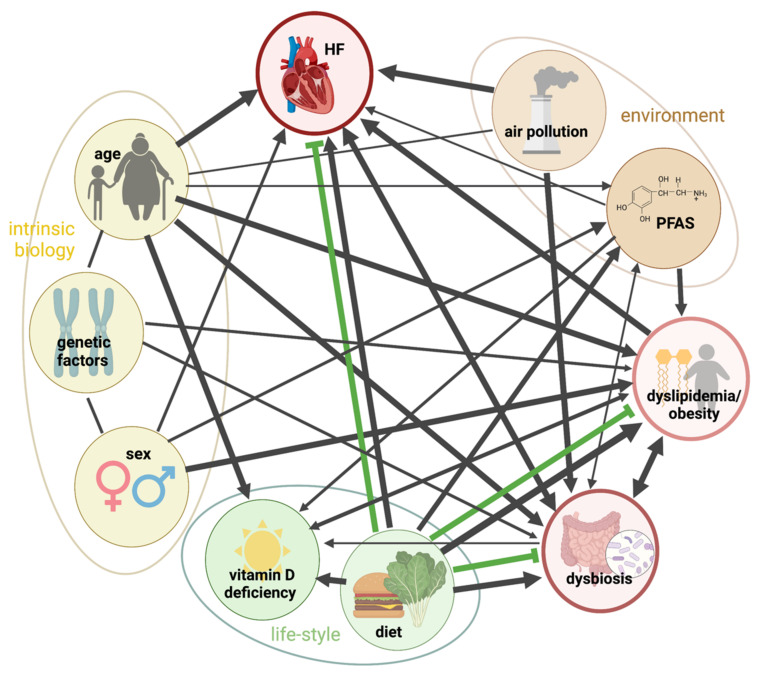
Interplay between the MFs investigated in this study. Black arrows: negative effect (arrow width indicating approximative magnitude). Green inhibitor arrows: potential positive effect. Arrows in both directions: mutual effect. Created with *Biorender.com*.

**Table 1 jcm-11-04464-t001:** Therapeutic intervention against COVID-19.

Clinical Trials				
**Therapeutic Category**	**Drug** **Generic Name** **(Trade Name)**	**Clinical Trial**	**Outcomes**	**Reference**
**IL-6 receptor inhibitors (monoclonal antibody)**	TOCILIZUMAB (Actemra/RoActemra)	RECOVERY (NCT04381936)	621 patients (31%) who received Tocilizumab died within 28 days compared with 729 patients (35%) who received standard of care.	[106]
		EMPACTA (NCT04372186)	Tocilizumab reduced the need for mechanical ventilation in patients with COVID-19-associated pneumonia. 12% of patients receiving Tocilizumab received mechanical ventilation compared with 19.3% of patients in the placebo group (*p* = 0.04); however, did not improve rates of survival.	[107]
		REMAP-CAP (NCT02735707)	In critically ill patients with COVID-19 receiving organ support in ICUs, treatment with the IL-6 receptor antagonist tocilizumab improved outcomes, including survival.	[108,109]
**IL-6 receptor inhibitors (monoclonal antibody)**	SARILUMAB (Kevzara)	REMAP-CAP (NCT02735707)	In critically ill hospitalized adults receiving organ support in ICU mortality at Day 21 was 22.2% (10/45) for sarilumab, and 35.8% (142/397) for control. Results showed a median adjusted odds ratios in-hospital survival of 2.01 (95% credible interval, 1.18–4.71) compared with the control group.	[108]
**Janus kinase (JAK) inhibitor**	BARICITINIB (Olumiant)	The Phase 3 Adaptive COVID-19 Treatment Trial 2 (ACTT-2) (NCT04401579)	Olumiant plus Veklury (Remdesivir) significantly shortened median time to recovery from 8 days to 7 days compared with Veklury alone. Patients receiving Olumiant plus Veklury also had a significantly increased likelihood of better clinical status at 15 days and significantly fewer patients progressing to mechanical ventilation. The 28-day mortality was 5.1% in the combination group and 7.8% in the control group.	[110]
		The Phase 3 COV-BARRIER trial (NCT04421027)	Significantly fewer patients in the Olumiant group reached 60-day all-cause mortality compared with patients in the placebo group	[111]
		The Phase 3 COV-BARRIER trial (NCT04421027)	Baricitinib was the first immunomodulatory treatment to reduce COVID-19 mortality in a placebo-controlled trial	[112]
**Anticoagulant**	Heparin drugs: (UFH and LMWH)	REMAP-CAP, (NCT02735707) ACTIV-4a, (NCT04505774) ATTACC trial (NCT04372589)	Non-critically ill COVID-19 patients in the REMAP-CAP, ACTIV-4a, and ATTACC trials who were hospitalized for COVID-19 but not critically ill who received heparin were more likely to survive until being discharged or not need the use of supporting care compared with those who did not receive heparin.	[113]
		A Phase 3 trial, HEP-COVID, (NCT04401293)	A therapeutic dose of LMW heparin applied prophylactically reduced the risk of blood clots among patients hospitalized with COVID-19 with “very elevated D-dimer levels” compared with standard of care for thromboprophylaxis.	[114]
**Monoclonal antibodies cocktail**	Casirivimab/imdevimab (REGN-COV)	RECOVERY (NCT04381936)	For patients who had not mounted an antibody response on their own (seronegative), REGN-COV significantly reduced 28-day mortality compared with usual care, but there was no significant difference between patients who had already mounted an antibody response (seropositive) and usual care.	[115,116]
		A Phase 1/2/3 trial (NCT04425629)	The treatment resolved symptoms and reduced the SARS-CoV-2 viral load more rapidly than placebo and reduced the risk for any-cause hospitalization or mortality compared with a placebo group	[117,118]
		Phase 3 trial (NCT04452318)	Treatment with subcutaneous casirivimab and imdevimab significantly reduced the incidence of symptomatic COVID-19 among recently exposed, asymptomatic individuals.	[119]
**Monoclonal antibody**	SOTROVIMAB (Xevudy) *known as VIR-7831 and GSK4182136	The Phase 2/3 COMET-ICE trial (NCT04545060)	583 patients (291 in the sotrovimab group and 292 in the placebo group). Three patients (1%) who received sotrovimab progressed to severe disease that led to being hospitalized or dying compared with 21 patients (7%) who received a placebo.	[120]
**Antiviral**	REMDESIVIR (Veklury)	ACTT-1 trial (NCT04280705)	Remdesivir, compared to placebo, improved the time to recovery (from 15 days to a median of 11 days) in adults who were hospitalized with COVID-19 and had evidence of lower respiratory tract infection.	[121]
		CATCO trial (NCT04330690)	Hospitalized patients with COVID-19 treated with remdesivir had lower death rates (by about 4%) and 60-day mortality was 24.8% and 28.2%, respectively. Also had reduced need for oxygen and mechanical ventilation compared to people receiving standard-of-care treatments.	[122]
		A phase 3, randomized, open-label study and a real-world, retrospective, longitudinal cohort study (NCT04292899 and EUPAS34303)	Remdesivir was associated with significantly greater recovery and reduced odds of death compared with standard of care in patients with severe COVID-19. The recovery rate at Day 14 was higher in patients who received remdesivir (*n* = 312) compared with those who received standard of care (*n* = 818) (74.4% vs. 59%; *p* < 0.001). The mortality rate at Day 14 was also lower in the remdesivir group.	[123]
		PINETREE (NCT04501952)	In non-hospitalized patients who were at high risk for COVID-19 progression, a 3-day course of remdesivir demonstrated a statistically significant 87% reduction in risk for the composite primary endpoint of COVID-19 related hospitalization or all-cause death by day 28.	[124]
**Antiviral** **(Ribonucleoside analogue)**	MOLNUPIRAVIR (Lagevrio)	Phase 3 MOVe-OUT trial (NCT04575597)	Early treatment with molnupiravir reduced the risk of hospitalization or death by relative risk reduction 30% (6.8%, molnupiravir vs. 9.7%, placebo) (an absolute risk reduction of 3%) for non-hospitalized patients with mild or moderate COVID-19.	[125,126]
**Antiviral therapeutic combination** **(SARS-CoV-2 protease inhibitor)**	NIRMATRELVIR (former PF-07321332) + RITONAVIR (Paxlovid™)	Phase 2/3 high risk EPIC-HR trial (NCT04960202)	Patients who received Paxlovid showed an 89% reduction in hospitalization or death compared with placebo if the patient was treated within 3 days of developing symptoms and 88% if treated within five days of symptom onset. No deaths compared to placebo in non-hospitalized, high-risk adults with COVID-19.	[127,128,129]
		Phase 2/3 EPIC-SR (NCT05011513)	Showed 0.6% of patients were hospitalized compared with 2.4% in the placebo group, a 70% reduction in hospitalization and no deaths in the treated population.	[129,130]

**Table 2 jcm-11-04464-t002:** KERs and KEs mentioned in the text by their numbers. AOP-Wiki represents the central repository for AOP-related knowledge. Each KE and KER has an assigned number as a unique identifier. The KE pages describe “Key Events”, a measurable change in a biological system. KER pages explore the scientific evidence for causality between two KEs (i.e., event A leads to Event B). All https://aopwiki.org links were accessed on 17 March 2022.

Number	KER Name	Links to the AOP-Wiki
**Key Event Relationships related to initial viral infection**		
KER2056	Binding to ACE2 leads to increased viral entry	https://aopwiki.org/relationships/2056
KER2310	Increased viral entry leads to Increased SARS-CoV-2 production	https://aopwiki.org/relationships/2310
KER2311	Binding to ACE2 leads to ACE2 dysregulation	https://aopwiki.org/relationships/2311
**Key Event Relationships related to central inflammatory processes**		
KER2356	Hypofibrinolysis leads to increased proinflammatory mediators	https://aopwiki.org/relationships/2356
KER2303	TLR Dysregulation leads to increased proinflammatory mediators	https://aopwiki.org/relationships/2303
KER1703	Increased proinflammatory mediators lead to recruitment of inflammatory cells	https://aopwiki.org/relationships/1703
KER2354	Recruitment of inflammatory cells leads to hyper-inflammation	https://aopwiki.org/relationships/2354
KER2495	Intestinal hyperpermeability contributes to hyper-inflammation	https://aopwiki.org/relationships/2495

## Data Availability

Not applicable.

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
