# Peer review of "Factors Modulating COVID-19: A Mechanistic Understanding Based on the Adverse Outcome Pathway Framework"

_jcm, 2022, doi:10.3390/jcm11154464_

Round 1

Reviewer 1 Report

The authors presented the work "Factors modulating COVID-19: a mechanistic understanding based on the Adverse Outcome Pathway framework".

In general the work is well written and it is clear about the topic. 

Points of concern: https://www.nature.com/articles/s41598-022-13113-4

1. The aim of the study is not well established for the specific case of COVID-19 (authors refer to the aim of the general strategy but not for the pandemic). Please extend. 

2. Lines 108-109: yes, this is the concept of the epidemiological triad in infectious disease. Please refer to that. 

3. Line 118: Please refer to the relevance of the viral agent (the third element of the epidemiological triad): Cite works about the genomic surveillance of the SARS-CoV-2 and how mutations can also affect the pandemic. You are not focusing on that, but it is relevant to cite this aspect. In the same way, you can refer to reports of coinfections and the possible implications, including this work: 

https://www.nature.com/articles/s41598-022-13113-4 

The variants and lineages must be exploted in more detail from a clinical view.

4. Line 178. Please extend the comments with a cite of the work: https://link.springer.com/article/10.1007/s43657-022-00058-x  and talk about studies in which risk factors are distributed among the clinical profiles independently of the severity or symptomatology. 

5. Due to the diversity of aspects discussed, a summary table can be a recommended option for readers, including factors (in rows) and the events (columns) with the main topics in each case!

Author Response

Response to Reviewer 1 Comments

The authors presented the work "Factors modulating COVID-19: a mechanistic understanding based on the Adverse Outcome Pathway framework". In general the work is well written and it is clear about the topic. 

We thank the reviewer for his/her comments that we have addressed point by point here below.

Point 1. The aim of the study is not well established for the specific case of COVID-19 (authors refer to the aim of the general strategy but not for the pandemic). Please extend. 

Response 1: We modified the abstract to mention specifically the factors that were investigated here as those factors are modulating COVID-19 outcomes. We adapted the abstract to stay within 200 words.

Point 2. Lines 108-109: yes, this is the concept of the epidemiological triad in infectious disease. Please refer to that. 

Response 2: The reference to the concept of the epidemiological triad in infectious disease, which consists of an external agent, a susceptible host and an environment that brings the host and agent together, has been added in the text line 111.

Point 3. Line 118: Please refer to the relevance of the viral agent (the third element of the epidemiological triad): Cite works about the genomic surveillance of the SARS-CoV-2 and how mutations can also affect the pandemic. You are not focusing on that, but it is relevant to cite this aspect. In the same way, you can refer to reports of coinfections and the possible implications, including this work: https://www.nature.com/articles/s41598-022-13113-4 

The variants and lineages must be exploited in more detail from a clinical view.

Response 3: We added a discussion on the genomic surveillance of the SARS-CoV-2 and how mutations can affect the pandemic line 123-136 with new references, including the one mentioned (https://www.nature.com/articles/s41598-022-13113-4 ) that highlights the possibility to trace co-infections via genomic surveillance.

Point 4. Line 178. Please extend the comments with a cite of the work: https://link.springer.com/article/10.1007/s43657-022-00058-x  and talk about studies in which risk factors are distributed among the clinical profiles independently of the severity or symptomatology. 

Response 4: We discussed line 180 p5 the study mentioned at point 4 reporting that risk factors (and particularly host co-morbidities) are distributed among the clinical profiles independently of the severity of the symptoms.

Point 5. Due to the diversity of aspects discussed, a summary table can be a recommended option for readers, including factors (in rows) and the events (columns) with the main topics in each case!

Response 5: We thank the reviewer for this comment. We thought about that option initially when writing the manuscript but then decided not to provide a summary table. Indeed, when working on that exercise, we realized that we were losing all the detailed and necessary balance description of the modulating impact of each factor to each causal relationship. The complexity of the relations as well as the current status of  knowledge do not permit to summarize the main topics in each case, but rather need a detailed paragraph for each event to provide adequately the current evidence and uncertainties.

Reviewer 2 Report

In this review, Clerbaux LA et al. provide an extensive description of identified factors modulating COVID19 infection outcomes that are exceptionally variable between patients. Particularly, the authors apply the Adverse Outcome Pathway (AOP) framework to identify and describe biological sequences of events leading to adverse events. This aims to both highlight current knowledge gaps, to orientate future research and to identify factors of high-risks patients and improve health care. Authors thus provide a mechanistic overview of the role and interplay of intrinsic and extrinsic factors (such as age, sex, genetic factors, comorbidities -  dyslipidemia and obesity, preexisting heart failure, gut dysbiosis, vitamin D deficiency - diet, PFAS, pollution and therapeutic intervention against COVID19) on different biology levels or key events that  lead to COVID19 pathology, from the initial SARS-CoV2 binding to its receptor ACE2 to inflammatory response.

This review is extensively detailed, informative as it summarizes the current knowledge on potential factors leading to adverse health outcomes, which are essential to identify problematic patients rapidly, and to provide adequate health care. The manuscript is clear, with supporting schematics, and well structured.

I would suggest minor specific clarifications:

-       Line 422 to define DAMPs at first occurrence such like PAMPs is defined

-       Line 438 and 440, to define mtDNA and cfDNA at first occurrence

-       Line 446, correct the sentence ‘On the other hand, [316] reported…’. It looks like that the intent was to cite specifically the authors for this study.

-       Line 1133, correct the sentence ‘obesity and lipid status and may also’

Author Response

Response to Reviewer 2 Comments

In this review, Clerbaux LA et al. provide an extensive description of identified factors modulating COVID19 infection outcomes that are exceptionally variable between patients. Particularly, the authors apply the Adverse Outcome Pathway (AOP) framework to identify and describe biological sequences of events leading to adverse events. This aims to both highlight current knowledge gaps, to orientate future research and to identify factors of high-risks patients and improve health care. Authors thus provide a mechanistic overview of the role and interplay of intrinsic and extrinsic factors (such as age, sex, genetic factors, comorbidities -  dyslipidemia and obesity, preexisting heart failure, gut dysbiosis, vitamin D deficiency - diet, PFAS, pollution and therapeutic intervention against COVID19) on different biology levels or key events that  lead to COVID19 pathology, from the initial SARS-CoV2 binding to its receptor ACE2 to inflammatory response.

This review is extensively detailed, informative as it summarizes the current knowledge on potential factors leading to adverse health outcomes, which are essential to identify problematic patients rapidly, and to provide adequate health care. The manuscript is clear, with supporting schematics, and well structured.

We thank the reviewer for his/her comments that we have addressed point by point here below.

I would suggest minor specific clarifications:

-       Line 422 to define DAMPs at first occurrence such like PAMPs is defined

Response: The definition of DAMP (damage-associated molecular patterns) is given earlier in the text (lines 31-32 of page 12). However, we included it again on line 434 of page 21.

-       Line 438 and 440, to define mtDNA and cfDNA at first occurrence

Response: The definitions were added for mitochondrial DNA (mtDNA) and circulating free DNA (cfDNA) line 438 and line 440, respectively.

-       Line 446, correct the sentence ‘On the other hand, [316] reported…’. It looks like that the intent was to cite specifically the authors for this study.

Response: The reviewer is right, we added “Olivieri et al.” before the reference line 446 in the revised manuscript.

-       Line 1133, correct the sentence ‘obesity and lipid status and may also’

Response: We corrected the sentence line 1172 in the revised manuscript by deleting the unnecessary “and”.

Reviewer 3 Report

The article covers the factors influencing the course of COVID-19 infection, both clinical and genetic, as well as those related to COVID itself. It is written in detail, carefully analyzed and thought out. I believe that in its present form it is a perfect complement and systematization of the current knowledge in this field.

Author Response

Response to Reviewer 3 Comments

The article covers the factors influencing the course of COVID-19 infection, both clinical and genetic, as well as those related to COVID itself. It is written in detail, carefully analyzed and thought out. I believe that in its present form it is a perfect complement and systematization of the current knowledge in this field.

We thank the reviewers for their comments that we have addressed point by point. We also would like to mention that we made extra corrections to the manuscript that can be found in the revised version.

Point 1. Two email addresses were not correctly inserted and have been modified in the current version (in yellow page 1)

Point 2. The references were missing in chapter 4.3, we added all of them (11).

Point 3. We added in the acknowledgments that “the work was performed under the JRC Exploratory Research project CIAO – Modelling COVID-19 pathogenesis using the Adverse Outcome Pathway (AOP)”.